# Adaptation with Self-Evaluation to Improve Selective Prediction in LLMs

**Jiefeng Chen**[1*]  **Jinsung Yoon**[2]  **Sayna Ebrahimi**[2]
**Sercan Ö. Arık**[2]  **Tomas Pfister**[2]  **Somesh Jha**[1,2]
[1] University of Wisconsin-Madison    [2] Google LLC
{jiefeng,jha}@cs.wisc.edu
{jinsungyoon,saynae,soarik,tpfister}@google.com

## Abstract

Large language models (LLMs) have recently shown great advances in a variety of tasks, including natural language understanding and generation. However, their use in high-stakes decision-making scenarios is still limited due to the potential for errors. *Selective prediction* is a technique that can be used to improve the reliability of the LLMs by allowing them to abstain from making predictions when they are unsure of the answer. In this work, we propose a novel framework for adaptation with self-evaluation to improve the selective prediction performance of LLMs. Our framework is based on the idea of using parameter-efficient tuning to adapt the LLM to the specific task at hand while improving its ability to perform self-evaluation. We evaluate our method on a variety of question-answering (QA) datasets and show that it outperforms state-of-the-art selective prediction methods. For example, on the CoQA benchmark, our method improves the AUACC from 91.23% to 92.63% and improves the AUROC from 74.61% to 80.25%.

## 1  Introduction

Large Language Models (LLMs) have recently demonstrated impressive capabilities in many natural language understanding, reasoning and generation tasks, such as question answering (Jiang et al., 2021; Singhal et al., 2023), summarization (Tang et al., 2023; Zhang et al., 2023b), semantic classification, and code generation (Poesia et al., 2022; Zhang et al., 2023a). As LLMs improve their remarkable performance, they are being increasingly considered to replace humans to perform high-stakes tasks. For example, LLMs can be used for medical QA to assist patients (Singhal et al., 2022). However, LLMs are not guaranteed to be accurate for all queries, so it is important to understand which queries they are reliable for. This

---

*  Work done during internship at Google and before joining Amazon.

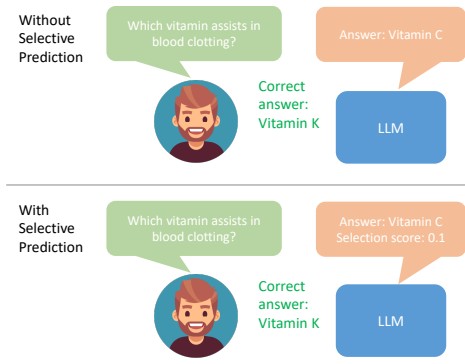

Figure 1: A safety-critical question from the TriviaQA dataset: "Which vitamin helps regulate blood clotting?" The OPT-2.7B model incorrectly answers "Vitamin C", when the correct answer is "Vitamin K". Without selective prediction, LLMs will directly output the wrong answer which in this case could lead users to take the wrong medicine, and thus causing potential harm. With selective prediction, LLMs will output a low selection score along with the wrong answer and can further output "I don't know!" to warn users not to trust it or verify it using other sources.

information can be used to direct human oversight to the queries with the lowest selection score. *Selective prediction* (Geifman and El-Yaniv, 2017), broadly refers to the deployment scenario for AI models where humans are involved to maintain overall accuracy by reviewing AI-generated, low-confidence outputs. In this scenario, both human and AI performance are considered together to minimize human involvement cost. LLMs should be used in the real-world with enhanced selective prediction performance. They should be able to assess the accuracy of their predictions and refrain from making wrong predictions. If an LLM detects that an answer might be wrong for a question, it should be able to generate an answer with the sentiment of "I don't know!" (as shown in Fig. 1) or defer the answer to a human for manual inspection. This will help to ensure that LLMs are used in a reliably, especially for high-stakes applications.

Selective prediction for LLMs is challenging because LLMs are just trained to predict the next to-

ken given a context but are not guaranteed to always predict the correct next token. Also, since LLMs generate an output sequence in an auto-regressive way, they don't directly produce a confidence score for the output sequence. Thus, obtaining selection scores from LLMs for their output sequences is not straightforward. Although there is some research on selective prediction for LLMs, these studies have their own shortcomings. Kadavath et al. propose to use heuristic prompts (e.g., adding prompts like "Is the proposed answer True or False?") to trigger self-evaluation of LLMs. However, those prompts may only work for the LLM used in Kadavath et al. (2022) and may not generalize to other types of LLMs (e.g., OPT and GPT2 models evaluated in our work). Some approaches proposed using semantic entropy (Kuhn et al., 2023) or self-consistency (Wang et al., 2022) as a measure of uncertainty for selection score. However, they usually require generating multiple output sequences to obtain the uncertainty measure for an input sequence, which introduces high computational cost and latency at test time. Fine-tuning LLMs on training data from the target question answering task using the standard LLM training loss can improve selective prediction performance. This is because fine-tuning can improve the accuracy of the predictions and maximize the likelihood of the ground-truth answer for a given question. However, maximizing the likelihood of the ground-truth answer is not the same as minimizing the likelihood of the wrong answers, since LLMs generate output sequences in an auto-regressive way. Even after fine-tuning, some wrong answers may still have high likelihood and be generated by the LLM at test time. Therefore, distinguishing correct and incorrect answers based on likelihood scores alone is a challenging task.

To address these challenges of self-evaluation and uncertainty estimation, we propose a novel framework – *Adaptation with Self-Evaluation to Improve Selective Prediction in LLMs (ASPIRE)*. Unlike previous methods that rely on hand-crafted heuristics or multiple output sequences, our framework learns to self-evaluate from target-task data. We do this by training LLMs on a subset of the training data from the question-answering tasks. This allows the LLMs to learn to distinguish between correct and incorrect answers on their own. We then define a selection score that combines the likelihood of the generated answer with the learned self-eval score (see Eq. (11)) to make selective predictions. This makes our method much less computationally expensive than solutions that require generating multiple output sequences to obtain the uncertainty measure. Thus, the proposed method is useful for practical applications where high selective prediction performance and low inference costs are desired, after deploying the LLM. In such applications, practitioners prefer collecting some training data to fine-tune smaller LLMs to achieve high selective prediction performance rather than directly deploying very large pre-trained LLMs with limited selective prediction performance for specific tasks.

We conduct extensive experiments to evaluate our proposed framework, ASPIRE. We show that ASPIRE achieves the state-of-the-art selective prediction performance on three question answering datasets: CoQA, TriviaQA and SQuAD, using OPT and GPT-2 models. We also provide empirical analysis to delve deeper into our proposed technique.

## 2   Related Work

**Selective Prediction for LLMs.**   Recently, LLMs (e.g., GPT-4 (OpenAI, 2023) and PaLM (Chowdhery et al., 2022)) have achieved great success in solving various kinds of Natural Language Generation (NLG) tasks. However, LLMs are still not very reliable and may generate wrong outputs when solving NLG tasks. Due to this, selective prediction (or sometimes called selective generation (Ren et al., 2022)) is critical for safely deploying LLMs in the real-world. Different from selective prediction for classification tasks (e.g., Natural Language Inference (NLI) tasks) (Xin et al., 2021), selective prediction for LLMs in solving NLG tasks is fundamentally different since the prediction is done auto-regressively over many steps and the possible answer set has an infinite size. Recently, several work propose some uncertainty measures for LLMs, which can be used for selective prediction (Si et al., 2022; Kadavath et al., 2022; Varshney et al., 2022; Ren et al., 2022; Kuhn et al., 2023). Some recent work studies selective prediction for solving question answering tasks where questions are ambiguous (Cole et al., 2023; Yin et al., 2023). Varshney and Baral (2023) propose a selective prediction method that at inference time leverages an auxiliary model which is trained to distinguish the correct predictions of the QA model from the incorrect ones. Different from previous work, our

work proposes to improve selective prediction performance of LLMs in solving question answering tasks by learning self-evaluation during fine-tuning.

**Parameter Efficient Fine-tuning.** Fine-tuning pretrained LLMs on downstream datasets can bring huge performance gains when compared to using the pretrained LLMs out-of-the-box (e.g., k-shot inference). However, as LLMs get larger and larger, full fine-tuning becomes very expensive in terms of computational cost and memory requirements. In addition, massive models might not be data efficient and overfitting issues might be observed, yielding suboptimal generalization. To address these issues, Parameter-Efficient Fine-tuning (PEFT) approaches have been proposed. PEFT approaches only fine-tune a small number of (extra) model parameters while freezing most parameters of the pretrained LLMs, thereby greatly decreasing the computational and storage costs. It has also been shown that PEFT approaches are better than fine-tuning in the low-data regimes and generalize better to out-of-domain scenarios. Existing PEFT approaches include LoRA (Hu et al., 2021), Prefix Tuning (Liu et al., 2021a), Soft Prompt Tuning (Lester et al., 2021) and P-Tuning (Liu et al., 2021b). In this work, we use Soft Prompt Tuning to learn self-evaluation to improve selective prediction performance of LLMs.

## 3 Problem Setup

Suppose we have a pre-trained LLM $f$ for an arbitrary generative modeling task such as question answering. The output can be represented as a sequence of tokens from the vocabulary $\mathcal{V}$. Let $\mathcal{V}^*$ be the space of sequences of tokens. Suppose the logits of $f$ on $v \in \mathcal{V}$ given $\mathbf{x} \in \mathcal{V}^*$ is $\bar{f}(v \mid \mathbf{x})$. The likelihood of the next token following $\mathbf{x}$ being $v$ is defined as:

$$f(v \mid \mathbf{x}) := \frac{\exp\left(\bar{f}(v \mid \mathbf{x})\right)}{\sum_{v' \in \mathcal{V}} \exp\left(\bar{f}(v' \mid \mathbf{x})\right)}, \quad (1)$$

whereas the likelihood of generating $\hat{\mathbf{y}} \in \mathcal{V}^*$ given $\mathbf{x}$ is defined as:

$$f(\hat{\mathbf{y}} \mid \mathbf{x}) := \Pi_{i=1}^{|\hat{\mathbf{y}}|} f(\hat{y}_i \mid \mathbf{x}, \hat{y}_{[i-1]}), \quad (2)$$

where $\hat{\mathbf{y}} = (\hat{y}_1, \ldots, \hat{y}_{|\hat{\mathbf{y}}|})$, $|\hat{\mathbf{y}}|$ is the length of $\hat{\mathbf{y}}$, $\hat{y}_{[i-1]} = (\hat{y}_1, \ldots, \hat{y}_{i-1})$ for $i > 0$ and $\hat{y}_{[0]} = \emptyset$. This likelihood can be very small when $|\hat{\mathbf{y}}|$ is very large. To address this issue, we define the normalized likelihood as:

$$f_{\text{norm}}(\hat{\mathbf{y}} \mid \mathbf{x}) := f(\hat{\mathbf{y}} \mid \mathbf{x})^{\frac{1}{|\hat{\mathbf{y}}|}} \quad (3)$$

We use $f$ to generate the output sequence for the given input $\mathbf{x}$ by solving the following objective:

$$\hat{\mathbf{y}}^* = \underset{\hat{\mathbf{y}}}{\operatorname{argmax}} \log f(\hat{\mathbf{y}} \mid \mathbf{x}) \quad (4)$$

It is impossible to solve this objective exactly since the output sequences can be arbitrarily long. However, we can employ some decoding strategy like greedy decoding or beam search to solve it.

To evaluate if the generated output $\hat{\mathbf{y}}$ is correct or not, we need a set of reference outputs $S$ and an evaluation metric $M : \mathcal{V}^* \times \mathcal{V}^* \to [0, 1]$ that can evaluate the similarity of the generated output $\hat{\mathbf{y}}$ compared to the reference output $\mathbf{y}_r \in S$. With a threshold $\gamma$, we can determine the correctness of the generated output – if $\max_{\mathbf{y}_r \in S} M(\hat{\mathbf{y}}, \mathbf{y}_r) > \gamma$, then the generated output is correct; otherwise, the generated output is wrong. We discuss the specific choices of $M$ and $\gamma$ in Section 6.

In selective prediction, we need a rejection option, which is denoted by $\perp$. Given a training dataset $\mathcal{D}^{tr} = \{(\mathbf{x}^i, \mathbf{y}^i)\}_{i=1}^{n_{tr}}$ randomly sampled from a target task distribution, we aim to build a selective predictor $f_s : \mathcal{V}^* \to \mathcal{V}^* \cup \{\perp\}$ that can achieve strong selective prediction performance on the test dataset $\mathcal{D}^{te} = \{(\mathbf{x}^i, S^i)\}_{i=1}^{n_{te}}$, where $S^i$ is the set of reference outputs for the input $\mathbf{x}^i$. The selective predictor $f_s$ is composed of a predictor $\hat{f} : \mathcal{V}^* \to \mathcal{V}^*$ and a selection scoring function $g : \mathcal{V}^* \to \mathbb{R}$. With $\hat{f}$ and $g$, the selective predictor $f_s$ is proposed as:

$$f_s(\mathbf{x}; \tau) = \begin{cases} \hat{f}(\mathbf{x}) & \text{if } g(\mathbf{x}) \geq \tau, \\ \perp & \text{if } g(\mathbf{x}) < \tau \end{cases}, \quad (5)$$

where $\tau$ is a threshold. The accuracy of the selective predictor is defined as the fraction of the accepted inputs where the predictions are correct. The coverage of the selective predictor is defined as the fraction of the inputs that are accepted. We can tune the threshold $\tau$ to achieve a certain coverage and there would be an accuracy-coverage trade-off.

We use the area under the accuracy-coverage curve (AUACC) metric to measure selective prediction performance and use the area under the receiver operator characteristic curve (AUROC) metric to measure the quality of the selection score estimation. AUACC is the common metric used for evaluating selective prediction performance (Xin et al., 2021; Yoshikawa and Okazaki, 2023). AUROC is equivalent to the probability that a randomly chosen correct output sequence has a higher

selection score than a randomly chosen incorrect output sequence. AUROC is used in (Kuhn et al., 2023) for evaluating uncertainty estimation methods.

## 4 ASPIRE Framework

We propose that LLMs should have the self-evaluation ability such that they should be able to distinguish whether their proposed answers for a given question are correct or not. Although some previous work (Kadavath et al., 2022) show that LLMs have good self-evaluation ability with specially designed prompts, those prompts may not transfer to different kinds of LLMs (as shown by our experiments and in Kuhn et al. (2023)) and hand-crafting prompts for different kinds of LLMs can be expensive. A more effective approach is to collect some training data to employ self-evaluation. Towards this end, we propose a novel framework – Adaptation with Self-Evaluation to Improve Selective Prediction in LLMs (ASPIRE). Fig. 2 illustrates the proposed framework and the details are explained next.

Given a training dataset for a generative task, we can fine-tune the pre-trained LLM on the training data to improve its prediction performance. Towards this end, parameter efficient tuning techniques (e.g., soft prompt tuning (Lester et al., 2021) and LoRA (Hu et al., 2021)) might be employed to adapt the pre-trained LLM on the task, given their effectiveness in obtaining strong generalization with small amount of target task data. Specifically, the model parameters $\theta$ of the LLM are frozen and adaptable parameters $\theta_p$ are added for fine-tuning. Only $\theta_p$ are updated to solve the following training objective:

$$\min_{\theta_p} \mathbb{E}_{(\mathbf{x},\mathbf{y}) \sim \mathcal{D}^{tr}} \mathcal{L}(\mathbf{x}, \mathbf{y}; \theta, \theta_p), \qquad (6)$$

where $\mathcal{L}$ is the LLM training loss (e.g. cross-entropy). Such fine-tuning can improve selective prediction performance because it not only improves the prediction accuracy, but also enhances the likelihood of correct output sequences.

To further improve selective prediction performance, we propose to fine-tune the LLM to learn self-evaluation. We first use the LLM with the learned $\theta_p$ to generate different answers for each example $(\mathbf{x}, \mathbf{y}) \in \mathcal{D}^{tr}$. Suppose the decoding algorithm used to generate output sequences for each input $\mathbf{x}$ is $\mathcal{A}$. $\mathcal{A}$ would produce a list of generated output sequences:

$$\mathcal{A}(f, \theta_p, \mathbf{x}) = [\hat{\mathbf{y}}^1, \dots, \hat{\mathbf{y}}^k], \qquad (7)$$

where $k$ is the number of output sequences generated. We aim to generate output sequences that have high likelihood (i.e., $f(\hat{\mathbf{y}}^j \mid \mathbf{x}; \theta_p)$ is high). We use the metric $M$ defined in Section 3 to determine if the generated output $\hat{\mathbf{y}}^j$ is correct or not. If $M(\hat{\mathbf{y}}^j, \mathbf{y}) > \hat{\gamma}$, we label $\hat{\mathbf{y}}^j$ as a correct output for $\mathbf{x}$; otherwise, we label $\hat{\mathbf{y}}^j$ as a wrong output for $\mathbf{x}$. Here, the threshold $\hat{\gamma}$ might be different from the threshold $\gamma$ used for evaluation. We choose a sufficiently large value of $\hat{\gamma}$ (e.g., $\hat{\gamma} = 0.9$) so that the generated wrong outputs wouldn't be labeled as correct outputs. In Appendix H, we provide more details and analyses on selection of $\hat{\gamma}$.

After sampling high-likelihood outputs for each query, we add adaptable parameters $\theta_s$ and only tune $\theta_s$ for learning self-evaluation. Since the output sequence generation only depends on $\theta$ and $\theta_p$, freezing $\theta$ and the learned $\theta_p$ can avoid changing the prediction behaviors of the LLM when learning self-evaluation. Let $z_c$ and $z_w$ be a pair of tokens that represent the words "correct" and "wrong" respectively. We can then optimize $\theta_s$ using the following training objective:

$$\begin{aligned} \min_{\theta_s} \quad & \mathbb{E}_{(\mathbf{x},\mathbf{y}) \sim \mathcal{D}^{tr}} \quad \mathcal{L}_c + \mathcal{L}_w \\ \mathcal{L}_c = & \mathbb{E}_{\hat{\mathbf{y}} \sim S_c(\mathbf{x},\mathbf{y})} - \log f(z_c | \mathbf{x}, \hat{\mathbf{y}}; \theta_p, \theta_s) \\ \mathcal{L}_w = & \mathbb{E}_{\hat{\mathbf{y}} \sim S_w(\mathbf{x},\mathbf{y})} - \log f(z_w | \mathbf{x}, \hat{\mathbf{y}}; \theta_p, \theta_s) \end{aligned}$$
$$(8)$$

where $S_c(\mathbf{x}, \mathbf{y})$ is a set of correct outputs containing the reference output $\mathbf{y}$ and $k_c$ correct outputs with highest likelihood from $\mathcal{A}(f, \theta_p, \mathbf{x})$, and $S_w(\mathbf{x}, \mathbf{y})$ is a set of wrong outputs containing $k_w$ wrong outputs with highest likelihood from $\mathcal{A}(f, \theta_p, \mathbf{x})$. If $\mathcal{A}(f, \theta_p, \mathbf{x})$ has less than $k_c$ correct outputs (or has less than $k_w$ wrong outputs), we include all its correct outputs (or all its wrong outputs) in $S_c$ (or $S_w$). We ensure that $S_w$ contains at least one wrong output. If $\mathcal{A}(f, \theta_p, \mathbf{x})$ doesn't contain wrong outputs, we add a default wrong output (e.g., the empty string) to $S_w$.

After training $\theta_p$ and $\theta_s$, we obtain the prediction for the query $\mathbf{x}$ via solving the following objective:

$$\hat{\mathbf{y}}^* = \operatorname*{argmax}_{\hat{\mathbf{y}}} \log f(\hat{\mathbf{y}} \mid \mathbf{x}; \theta_p). \qquad (9)$$

We use the beam search decoding method towards this. We define the likelihood of the output $\hat{\mathbf{y}}^*$

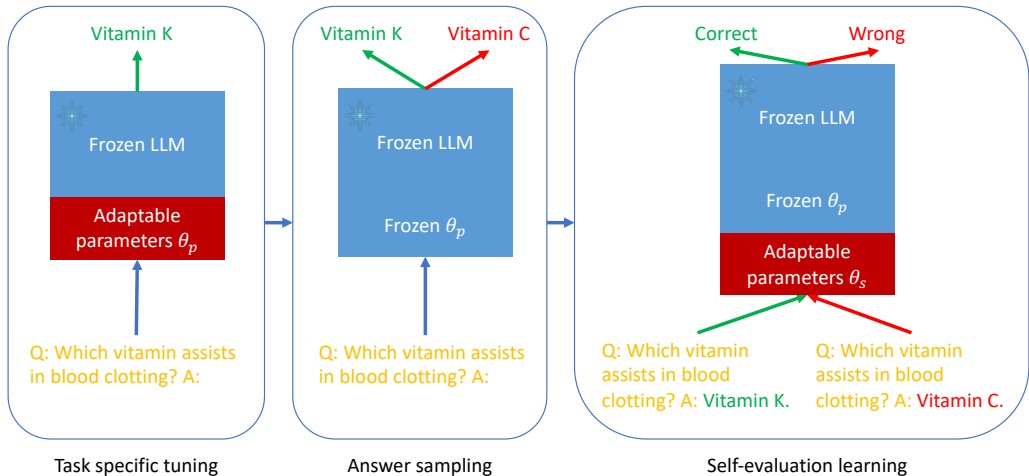

Figure 2: In the proposed framework ASPIRE, we first perform task specific tuning to train adaptable parameters $\theta_p$ while freezing the LLM. Then we use the LLM with the learned $\theta_p$ to generate different answers for each training question to create a dataset for self-evaluation learning. Finally, we train the adaptable parameters $\theta_s$ to learn self-evaluation using the created dataset while freezing the LLM and the learned $\theta_p$.

being correct for the query $\mathbf{x}$ as:

$$
P(z_c \mid \mathbf{x}, \hat{\mathbf{y}}^*) = \\
\frac{\exp\left(\bar{f}(z_c \mid \mathbf{x}, \hat{\mathbf{y}}^*; \theta_p, \theta_s)\right)}{\sum_{z \in \{z_c, z_w\}} \exp\left(\bar{f}(z \mid \mathbf{x}, \hat{\mathbf{y}}^*; \theta_p, \theta_s)\right)} \quad (10)
$$

This score $P(z_c \mid \mathbf{x}, \hat{\mathbf{y}}^*)$ is referred as the learned self-eval score. Overall, the selection scoring function is proposed as:

$$
g(\mathbf{x}) = (1 - \alpha) \cdot \log f_{\text{norm}}(\hat{\mathbf{y}}^* \mid \mathbf{x}; \theta_p) \quad (11) \\
+ \alpha \cdot \log P(z_c \mid \mathbf{x}, \hat{\mathbf{y}}^*).
$$

where $\alpha \in [0, 1]$ is a hyper-parameter.

## 5 Implementation via Soft Prompt Tuning

In the proposed framework, $\theta_p$ and $\theta_s$ can be trained using parameter efficient tuning approaches. In our work, we focus on Soft Prompt Tuning, as illustrated in Fig. 3. The driving force behind this approach lies in the recognition that if we can develop prompts that effectively stimulate self-evaluation, it should be possible to discover these prompts through soft prompt tuning in conjunction with targeted training objectives.

We first briefly introduce the soft prompt tuning method proposed by Lester et al. (2021). We consider LLMs based on the Transformer architecture (Vaswani et al., 2017). Given a query $\mathbf{x} = (x_1, \ldots, x_{m_q})$, Transformers first embed the tokens, forming a matrix $X \in \mathbb{R}^{m_q \times d_e}$, where $d_e$ is the dimension of the embedding space. The soft-prompts are represented as parameters $\tilde{\theta} \in \mathbb{R}^{l \times d_e}$,

where $l$ is the length of the prompt. The prompt is then concatenated to the embedded input forming a single matrix $[\tilde{\theta}; X] \in \mathbb{R}^{(m_q+l) \times d_e}$, which then flows through the transformer as normal.

In the proposed framework, we need to train two portions of the prompts $\theta_p \in \mathbb{R}^{l \times d_e}$ and $\theta_s \in \mathbb{R}^{l \times d_e}$. Utilizing soft prompt tuning, the training objective (6) is proposed as:

$$
\min_{\theta_p} \mathbb{E}_{(\mathbf{x}, \mathbf{y}) \sim \mathcal{D}^{tr}} \frac{1}{|\mathbf{y}|} \sum_{j=1}^{|\mathbf{y}|} -\log f(y_j \mid [\theta_p; X; Y_{[j-1]}]),
$$
$$(12)$$

where $X$ is the embedding of $\mathbf{x}$ and $Y_{[j-1]}$ is the embedding of $y_{[j-1]}$. On the other hand, the training objective (8) is proposed as:

$$
\min_{\theta_s} \quad \mathbb{E}_{(\mathbf{x}, \mathbf{y}) \sim \mathcal{D}^{tr}} \quad \mathcal{L}_c + \mathcal{L}_w \\
\mathcal{L}_c = \mathbb{E}_{\hat{\mathbf{y}} \sim S_c(\mathbf{x}, \mathbf{y})} - \log f(z_c \mid [\theta_p; X; \hat{Y}; \theta_s]) \\
\mathcal{L}_w = \mathbb{E}_{\hat{\mathbf{y}} \sim S_w(\mathbf{x}, \mathbf{y})} - \log f(z_w \mid [\theta_p; X; \hat{Y}; \theta_s])
$$
$$(13)$$

where $\hat{Y}$ is the embedding of $\hat{\mathbf{y}}$. The inference objective (9) in the framework becomes:

$$
\hat{\mathbf{y}}^* = \underset{\hat{\mathbf{y}}}{\arg\max} \log f(\hat{\mathbf{y}} \mid [\theta_p; X]) \quad (14)
$$

The learned self-eval score $P(z_c \mid \mathbf{x}, \hat{\mathbf{y}}^*)$ becomes:

$$
P(z_c \mid \mathbf{x}, \hat{\mathbf{y}}^*) = \\
\frac{\exp\left(\bar{f}(z_c \mid [\theta_p; X; \hat{Y}^*; \theta_s])\right)}{\sum_{z \in \{z_c, z_w\}} \exp\left(\bar{f}(z \mid [\theta_p; X; \hat{Y}^*; \theta_s])\right)} \quad (15)
$$

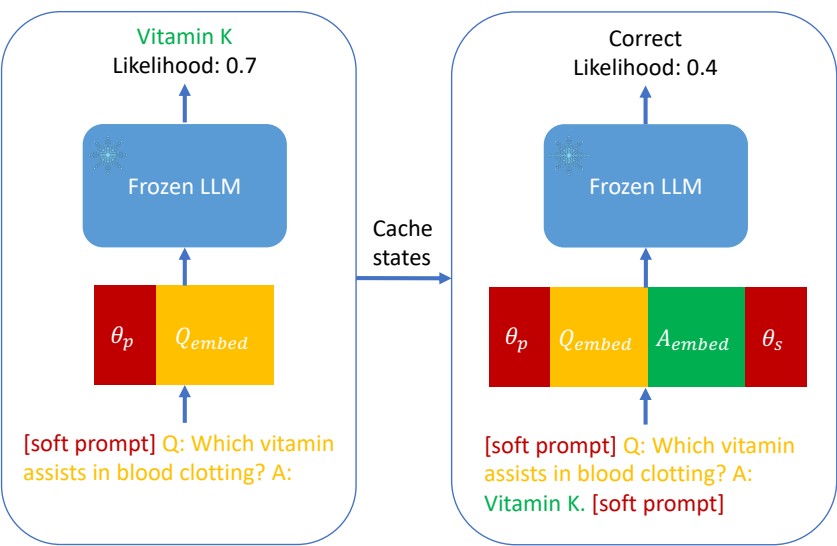

Figure 3: Implementation of the proposed framework via soft prompt tuning. $\theta_p$ and $\theta_s$ are learnable soft prompt embeddings. $Q_{embed}$ and $A_{embed}$ are input embeddings for the question and answer respectively. We first generate the answer and the likelihood of the answer, and then compute the learned self-eval score. We can cache the states when generating the answer and reuse those states when computing the learned self-eval score to save computational costs.

where $\hat{Y}^*$ is the embedding of $\hat{\mathbf{y}}^*$.

To generate the output sequence and obtain the selection score for a given input sequence, we employ two stages: first, we obtain the generated output and the likelihood for the generated output and then, we obtain the learned self-eval score. Since the query of the second stage is constructed by appending some additional tokens to the query of the first stage, the second stage can reuse the states in the first stage instead of recomputing them to save some computational cost (see Fig. 3).

Lastly, we note that the computational complexity of the proposed method at test time is $O(l_{max})$ with $l_{max}$ being the maximum length of the generated output sequence. In Appendix F, we provide a more detailed analysis of the computational complexity of different methods. The predictive entropy and semantic entropy methods have a complexity of $O(m \cdot l_{max})$ where $m$ is the number of output sequences sampled for uncertainty estimation, which is much larger than that of our method.

## 6 Experiments

Our experimental evaluation is focused on the following questions:

**(Q1)** Could a learning-based system using self-evaluation improve selective prediction in LLMs compared to other post-hoc selective prediction alternatives?

**(A1)** By learning self-evaluation, we can significantly improve selective prediction performance

across different datasets and LLMs (see Table 1).

**(Q2)** What kinds of decoding algorithms could be used as $\mathcal{A}$ for the proposed framework ASPIRE?

**(A2)** Using decoding algorithms that can sample different high-likelihood answers as $\mathcal{A}$ (e.g., beam search) is important for ASPIRE to achieve good selective prediction performance (see Table 4).

**(Q3)** What is the effect of the number of training samples for the proposed method ASPIRE?

**(A3)** More training samples lead to enhanced performance and with ~2k samples, ASPIRE can outperform the baselines without soft prompt tuning significantly on different datasets (see Table 5).

### 6.1 Setup

**Dataset.** We focus on the free-form question answering tasks on the datasets CoQA (Reddy et al., 2019), TriviaQA (Joshi et al., 2017) and SQuAD (Rajpurkar et al., 2016). For CoQA and SQuAD, since each question is asked based on a context paragraph, we evaluate the LLMs in the zero-shot setting. For TriviaQA, since the LLMs have limited accuracy under the zero-shot setting, we evaluate the LLMs in 5-shot setting. For each dataset, we use a subset of the original training set containing 50K examples for adapting LLMs by default. The details of the datasets are given in Appendix B.

**LLMs.** We use OPT (Zhang et al., 2022) and GPT-2 (Radford et al., 2019) models of various sizes. For OPT, we consider OPT-350M, OPT-1.3B,

| Model | Method | CoQA | | TriviaQA | | SQuAD | |
|---|---|---|---|---|---|---|---|
| | | AUACC ↑ | AUROC ↑ | AUACC ↑ | AUROC ↑ | AUACC ↑ | AUROC ↑ |
| Pre-trained GPT2-XL | Perplexity | 55.93 | 62.05 | 22.60 | 72.88 | 7.68 | 51.90 |
| | Predictive Entropy | 60.76 | 67.53 | 24.83 | 76.20 | 10.04 | 57.21 |
| | Semantic Entropy | 63.03 | 70.50 | 24.37 | 75.33 | 10.38 | 59.17 |
| | Self-eval | 46.67 | 50.83 | 9.30 | 42.75 | 7.32 | 49.56 |
| | P(True) | 46.98 | 51.17 | 10.62 | 44.54 | 10.69 | 60.87 |
| Adapted GPT2-XL with $\theta_p$ | Perplexity | 83.27 | 72.79 | 36.49 | 79.92 | 88.73 | 75.08 |
| | Predictive Entropy | 83.49 | 73.44 | 37.31 | 82.21 | 88.25 | 74.16 |
| | Semantic Entropy | 84.40 | 75.16 | 36.68 | 81.40 | 88.62 | 75.26 |
| | Self-eval | 69.91 | 51.90 | 14.39 | 43.33 | 74.26 | 49.13 |
| | P(True) | 70.63 | 52.83 | 13.59 | 40.59 | 74.34 | 49.09 |
| | ASPIRE (ours) | **85.65** | **78.32** | **38.06** | **83.23** | **89.86** | **78.35** |
| Pre-trained OPT-2.7B | Perplexity | 75.26 | 70.16 | 40.93 | 78.86 | 40.82 | 57.20 |
| | Predictive Entropy | 75.29 | 69.16 | 41.20 | 78.92 | 47.18 | 62.85 |
| | Semantic Entropy | 76.31 | 70.96 | 40.72 | 78.06 | 51.53 | 68.40 |
| | Self-eval | 62.32 | 52.26 | 25.88 | 59.04 | 41.78 | 59.05 |
| | P(True) | 62.16 | 51.80 | 24.88 | 56.89 | 34.77 | 49.42 |
| Pre-trained OPT-30B | Self-eval | 71.99 | 51.10 | 36.92 | 48.90 | 46.24 | 57.26 |
| | P(True) | 71.59 | 51.31 | 36.20 | 45.63 | 43.93 | 54.26 |
| Adapted OPT-2.7B with $\theta_p$ | Perplexity | 90.80 | 74.23 | 53.56 | 81.74 | 92.86 | 75.72 |
| | Predictive Entropy | 90.63 | 72.87 | 53.91 | 82.19 | 92.96 | 75.58 |
| | Semantic Entropy | 91.23 | 74.61 | 53.58 | 81.55 | 93.21 | 76.53 |
| | Self-eval | 81.30 | 50.76 | 32.98 | 56.03 | 86.34 | 56.99 |
| | P(True) | 81.14 | 51.01 | 33.48 | 56.27 | 82.59 | 49.48 |
| | ASPIRE (ours) | **92.63** | **80.25** | **55.06** | **84.44** | **94.73** | **82.60** |

Table 1: Results of evaluating different methods to compute the selection score when the model's predictions are fixed. All numbers are percentages. **Bold** numbers are superior results.

OPT-2.7B and OPT-30B. For GPT-2, we consider GPT2-Medium, GPT2-Large and GPT2-XL. The details of these models are given in Appendix C.

**Baselines.** For selective prediction, we need to get a predicted output sequence $\hat{\mathbf{y}}^*$ and a selection score $g(\mathbf{x})$ for each input sequence $\mathbf{x}$ given a model $f$. The model $f$ can be a pre-trained LLM or an adapted LLM with $\theta_p$ trained using the training objective (12). We use the beam-search decoding to obtain the predicted output sequence $\hat{\mathbf{y}}^*$ and consider the following baselines to compute the selection score $g(\mathbf{x})$: (1) Perplexity; (2) Predictive Entropy; (3) Semantic Entropy (Kuhn et al., 2023); (4) Self-eval; (5) P(True) (Kadavath et al., 2022). More details can be found in Appendix D.

**Evaluation metrics.** We use the Rouge-L (Lin and Och, 2004) as the evaluation metric $M$ to evaluate the similarity of the generated answer to the reference answers following Kuhn et al. (2023). For the threshold $\gamma$ that is used to determine the correctness of the generated answer, we consider relatively larger values of $\gamma$ since we focus on safety-critical applications where accepting a wrong answer is more costly compared to rejecting a correct answer

that is different from the reference answers (refer to Appendix G for the justifications of the choices of $\gamma$). Unless specified, we use $\gamma = 0.7$ as default.

**Training hyper-parameters.** We have two stages of training: the first stage is to train the soft prompt $\theta_p$ using the training objective (12) and the second stage is to train the soft prompt $\theta_s$ using the training objective (13). For both stages, we train the soft prompts for 10 epochs using AdamW optimizer with a batch size of 8, a learning rate of 0.01 and cosine learning rate scheduling. More training details can be found in Appendix E.

**ASPIRE setup.** We use the beam search as the decoding algorithm $\mathcal{A}$. We set the number of beams equal to $k$ and use the $k$ highest scoring beams as the answer list $\mathcal{A}(f, \theta_p, \mathbf{x})$. We set $l = 50$, $\hat{\gamma} = 0.9$, $k = 10$, $k_c = 2$, $k_w = 10$ and $\alpha = 0.25$ by default. We choose these hyper-parameters based on the performance on the validation set from TriviaQA using the OPT-2.7B model. We then use the same hyper-parameters across all datasets and models.

| Model | Method | CoQA | | TriviaQA | | SQuAD | |
|---|---|---|---|---|---|---|---|
| | | AUACC ↑ | AUROC ↑ | AUACC ↑ | AUROC ↑ | AUACC ↑ | AUROC ↑ |
| Adapted OPT-2.7B with $\theta_p$ | ASPIRE ($\alpha = 0.0$) | 90.80 | 74.23 | 53.56 | 81.74 | 92.86 | 75.72 |
| | ASPIRE ($\alpha = 0.25$) | **92.63** | **80.25** | **55.06** | **84.44** | **94.73** | **82.60** |
| | ASPIRE ($\alpha = 0.5$) | 92.56 | 80.18 | 54.61 | 84.33 | 94.59 | 82.16 |
| | ASPIRE ($\alpha = 0.75$) | 92.05 | 78.37 | 52.71 | 81.52 | 94.28 | 80.98 |
| | ASPIRE ($\alpha = 1.0$) | 91.33 | 76.08 | 48.84 | 76.39 | 93.77 | 79.48 |

Table 2: Results of studying the effect of the hyper-parameter $\alpha$ in the proposed selection score (Eq. (11)). All numbers are percentages. **Bold** numbers are superior results.

| Model | CoQA | TriviaQA | SQuAD |
|---|---|---|---|
| | Acc ↑ | Acc ↑ | Acc ↑ |
| Pre-trained GPT2-XL | 46.27 | 11.80 | 7.41 |
| Adapted GPT2-XL with $\theta_p$ | 69.18 | 17.45 | 75.44 |
| Pre-trained OPT-2.7B | 60.68 | 21.38 | 35.95 |
| Pre-trained OPT-30B | 71.06 | 39.36 | 41.41 |
| Adapted OPT-2.7B with $\theta_p$ | 80.45 | 29.21 | 83.27 |

Table 3: Results of evaluating the accuracy of different LLMs. All numbers are percentages.

## 6.2 Results

We first evaluate the accuracy of different LLMs. The results in Table 3 show that after training $\theta_p$ via soft prompt tuning, the accuracy of LLMs is improved significantly. On the CoQA and SQuAD datasets, the adapted OPT-2.7B can even outperform the pre-trained OPT-30B, which demonstrates that it is possible to adapt a smaller LLM to achieve better accuracy than a much larger LLM. We then evaluate different methods to compute the selection score when the model's predictions are fixed. The results in Table 1 show that the proposed method ASPIRE significantly outperforms the baselines in terms of the AUACC and AUROC metrics across different datasets and LLMs. The results also show that after prompt tuning, the AUACC of different methods is significantly improved as the accuracy gets better and the perplexity becomes more meaningful in separating correct and wrong answers. Additionally, the results show that the proposed ASPIRE with the adapted OPT-2.7B model can significantly outperform the Self-eval and P(True) baselines with the pre-trained OPT-30B model in selective prediction performance. Note that on the TriviaQA dataset, although the pre-trained OPT-30B model has better accuracy than the adapted OPT-2.7B model, the Self-eval and P(True) baselines with the pre-trained OPT-30B model have much worse selective prediction performance compared to the proposed ASPIRE with the adapted

OPT-2.7B model. These demonstrate that the self-evaluation approaches are not effective for high capacity LLMs, and applying the proposed ASPIRE to smaller LLMs can lead to better selective prediction performance compared to those self-evaluation approaches with much larger LLMs. Additional results in Appendix I show that ASPIRE significantly outperforms the baselines across OPT and GPT2 models of different sizes for different values of the Rouge threshold $\gamma$.

## 6.3 Empirical Analyses

**The effect of $\alpha$.** We study the effect of the hyper-parameter $\alpha$ in the proposed selection score (Eq. (11)). The results in Table 2 show that setting $\alpha = 0.25$ leads to the best performance since it combines the normalized likelihood and the learned self-eval score in a good way. Only using the normalized likelihood (i.e., $\alpha = 0$) or only using the learned self-eval score (i.e., $\alpha = 1$) leads to much worse performance. In practice, the value of $\alpha$ can be chosen based on the performance on the validation data. In Appendix J, we give results for other models and discuss how we choose $\alpha$.

**The choices of $\mathcal{A}$.** We compare two decoding algorithms – beam search and multinomial sampling that can be used as $\mathcal{A}$ for answer sampling. For beam search, we use the $k$ highest scoring beams as the answer list. For multinomial sampling, we consider temperature (denoted as $T$) in the set $\{0.1, 1.0, 2.0\}$. The results in Table 4 show that using multinomial sampling with $T = 2.0$ or $T = 0.1$ leads to worse performance compared to other decoding algorithms. If we set a high temperature ($T = 2.0$) for multinomial sampling, then we sample some random answers that might not have high-likelihood. If we set a low temperature ($T = 0.1$) for multinomial sampling, then we repeatedly sample the same high-likelihood answers. Thus, the results suggest that sampling different high-likelihood answers is important for our

| Model | Decoding Algorithm | CoQA | | TriviaQA | | SQuAD | |
|---|---|---|---|---|---|---|---|
| | | AUACC ↑ | AUROC ↑ | AUACC ↑ | AUROC ↑ | AUACC ↑ | AUROC ↑ |
| Adapted GPT2-XL with $\theta_p$ | Multinomial (T=0.1) | 83.82 | 74.22 | 36.40 | 80.67 | 89.75 | 77.56 |
| | Multinomial (T=1.0) | 84.96 | 76.15 | 37.03 | 81.41 | **90.12** | **78.71** |
| | Multinomial (T=2.0) | 83.06 | 72.96 | 36.34 | 80.14 | 89.41 | 76.98 |
| | Beam search | **85.65** | **78.32** | **38.06** | **83.23** | 89.86 | 78.35 |
| Adapted OPT-2.7B with $\theta_p$ | Multinomial (T=0.1) | 92.04 | 77.96 | 55.09 | 84.28 | 94.24 | 80.52 |
| | Multinomial (T=1.0) | 92.60 | 79.86 | **55.15** | 84.29 | 94.57 | 82.08 |
| | Multinomial (T=2.0) | 92.02 | 77.91 | 53.80 | 82.40 | 94.15 | 80.42 |
| | Beam search | **92.63** | **80.25** | 55.06 | **84.44** | **94.73** | **82.60** |

Table 4: Results of comparing different decoding algorithms for answer sampling in the proposed method. We denote the temperature as $T$. All numbers are percentages. **Bold** numbers are superior results.

| Model | Method | CoQA | | TriviaQA | | SQuAD | |
|---|---|---|---|---|---|---|---|
| | | AUACC ↑ | AUROC ↑ | AUACC ↑ | AUROC ↑ | AUACC ↑ | AUROC ↑ |
| Pre-trained OPT-2.7B | Predictive Entropy | 75.29 | 69.16 | 41.20 | 78.92 | 47.18 | 62.85 |
| | Semantic Entropy | 76.31 | 70.96 | 40.72 | 78.06 | 51.53 | 68.40 |
| Adapted OPT-2.7B with $\theta_p$ | ASPIRE (size=1k) | 80.87 | 67.01 | 45.70 | 78.98 | 85.42 | 71.42 |
| | ASPIRE (size=2k) | 85.71 | 73.72 | 46.64 | 79.24 | 88.27 | 75.74 |
| | ASPIRE (size=5k) | 87.83 | 74.58 | 49.77 | 82.06 | 90.09 | 77.09 |
| | ASPIRE (size=10k) | 90.46 | 78.29 | 51.88 | 83.13 | 92.48 | 79.46 |
| | ASPIRE (size=50k) | 92.63 | 80.25 | 55.06 | 84.44 | 94.73 | 82.60 |

Table 5: Results of studying the effect of training set size for the proposed ASPIRE. All numbers are percentages.

method to achieve high accuracy and coverage in selective prediction. The results also show that using beam search leads to similar performance as using multinomial sampling with $T = 1$. So we can use either one in practice.

**Training sample efficiency.** We perform experiments to study the effect of the number of training samples for ASPIRE. We fix the number of training steps to be 50K while varying the size of the training dataset. The results in Table 5 show that more training samples lead to performance improvement and with 2K training samples, ASPIRE can outperform the baselines without soft prompt tuning by a large margin across different datasets. This underlines that our method, ASPIRE, can significantly improve selective prediction performance even with limited number of training samples.

## 7 Conclusion

In this paper, we proposed a novel framework for adaptation with self-evaluation to improve selective prediction in LLMs. We implemented the framework via soft prompt tuning and demonstrated its superior performance over existing methods through extensive experiments. In future work, one could explore implementing our framework via other parameter efficient tuning approaches and

applying our method to larger LLMs.

## Limitations

Higher capacity LLMs are known to often yield superior capabilities. Our work does not include fine-tuning experimental results with the largest and the strongest LLMs in the literature (we have fine-tuning results with LLMs up to 2.7B parameters), due to our computational constraints. However, the proposed framework can be applied to LLMs of any size and similar improvements are expected. We leave the adoption of our methods to larger-scale LLMs to future work.

## Ethics Statement

LLMs are widely used in various applications nowadays. However, they can generate wrong or misleading answers to questions, which can cause serious consequences in some safety critical applications. The framework proposed in our work can be used to improve selective prediction performance of LLMs and make their deployments more reliable. However, it is noted that the obtained selective prediction performances are still not perfect.

## Acknowledgements

We thank all the anonymous reviewers for their careful comments and feedback. The work is partially supported by Air Force Grant FA9550-18-1-0166, the National Science Foundation (NSF) Grants CCF-FMitF-1836978, IIS-2008559, SaTC-Frontiers1804648, CCF-2046710 and CCF-1652140, and ARO grant number W911NF-17-1-0405. Jiefeng Chen and Somesh Jha are partially supported by the DARPA-GARD problem under agreement number 885000.

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

## A  Hardware and Software

We run all experiments using the HuggingFace API on 40GB NVIDIA A100 GPUs in the Debian GNU/Linux 10 system. We use the OPT and GPT2 models via the HuggingFace transformers library which can be easily adapted for reproducibility. We modify the `Trainer` class provided by the HuggingFace API for soft prompt tuning. We use the `generate()` function of the HuggingFace API to generate answers. Unless specified, we use the default parameters of the `generate()` function. When generating the answer set $\mathcal{A}(f, \theta_p, \mathbf{x})$, we set `max_new_tokens=50` while in other cases, we always set `max_new_tokens=256`. The parameters for different decoding strategies are provided below:

- Beam search decoding: we set `num_beams>1` and `do_sample=False`. If we want to get `num_beams` highest scoring beams, we will set `num_return_sequences=num_beams`. We will specify `num_beams` when using beam search decoding.

- Multinomial sampling decoding: we set `num_beams=1` and `do_sample=True`. We will specify `temperature` when using multinomial sampling decoding.

## B  Datasets

We use three question answering datasets: CoQA (Reddy et al., 2019), TriviaQA (Joshi et al., 2017) and SQuAD (Rajpurkar et al., 2016) for experiments. The details about these datasets are given below.

### B.1  CoQA

CoQA is a large-scale dataset for Conversational Question Answering systems. The goal of the CoQA challenge is to measure the ability of machines to understand a text passage and answer a series of interconnected questions that appear in a conversation. CoQA contains 127,000+ questions with answers collected from 8,000+ conversations. The training set contains 108,647 question queries while the test set contains 7,983 question queries. We use the following template to construct question queries:

```
[The provided context paragraph]
[additional question-answer pairs]
Q: [Provided question]
A:
```

where additional question-answer pairs are preceding turns of the conversation about the paragraph consisting of questions and reference answers.

### B.2  TriviaQA

TriviaQA is a reading comprehension dataset containing over 650K question-answer-evidence triples. TriviaQA includes 95K question-answer pairs authored by trivia enthusiasts and independently gathered evidence documents, six per question on average, that provide high quality distant supervision for answering the questions. We focus on TriviaQA as a closed-book QA task (in which the model must answer a question without access to a supporting paragraph). The training set contains 138,384 question queries while the test set contains 17,944 question queries. We split the original test set into a new test set containing 8,000 question queries and a validation set containing 9,944 question queries. We use the new test set for evaluation and use the validation set for hyper-parameter selection. We consider the following template with a 5-shot prompt to construct question queries:

```
Q: In which decade did Billboard magazine
first publish and American hit chart? A:
30s. Q: What is Bruce Willis' real first
name? A: Walter. Q: Which city does David
Soul come from? A: Chicago. Q: Which
William wrote the novel Lord Of The Flies?
A: Golding. Q: Where in England was Dame
Judi Dench born? A: York. Q: [Provided
question] A:
```

### B.3  SQuAD

Stanford Question Answering Dataset (SQuAD) is a reading comprehension dataset, consisting of questions posed by crowd-workers on a set of Wikipedia articles, where the answer to every question is a segment of text, or span, from the corresponding reading passage. We use the SQuAD 1.1 version, containing 100,000+ question-answer pairs on 500+ articles. The training set contains 86,821 question queries while the test set contains 5,928 question queries. We use the following template to construct question queries:

```
[The provided context paragraph]
Q: [Provided question]
A:
```

## C  LLMs

We perform experiments with OPT (Zhang et al., 2022) and GPT-2 (Radford et al., 2019) models, which are based on Transformer architecture. For Transformer architecture, there is a limit on the lengths of the sequences we can pass the models. The OPT models can handle sequences of up to 2,048 tokens while the GPT-2 models can handle sequences of up to 1,024 tokens. If the sequence length of an input is larger than the maximum sequence length that is allowed, we force the model to output an empty sequence with a $-\infty$ selection score.

## D  Baselines

For selective prediction, we need to get a predicted output sequence $\hat{\mathbf{y}}^*$ and a selection score $g(\mathbf{x})$ for each input sequence $\mathbf{x}$ given a model $f$. The model $f$ can be a pre-trained LLM or an LLM adapted with prompt tuning using training objective (12). We use the beam-search decoding, with the number of beams being equal to 5, to obtain the predicted output sequence $\hat{\mathbf{y}}^*$. We consider the following baselines to compute the selection score $g(\mathbf{x})$:

**Perplexity.**  Perplexity is defined as the exponentiated average negative log-likelihood of a sequence. The perplexity of the generated output sequence $\hat{\mathbf{y}}^*$ is computed as:

$$\text{perp}(\hat{\mathbf{y}}^* \mid \mathbf{x}; f) = f_{\text{norm}}(\hat{\mathbf{y}}^* \mid \mathbf{x})^{-1} \qquad (16)$$

**Predictive Entropy.**  Predictive entropy is a widely used measure of uncertainty. We use the multinomial sampling with a temperature of $0.5$ to obtain an answer list $[\hat{\mathbf{y}}^1, \ldots, \hat{\mathbf{y}}^m]$ for each input sequence $\mathbf{x}$. The predictive entropy is computed as:

$$\text{pe}(\mathbf{x}; f) = \sum_{j=1}^{m} \frac{1}{m} \log f_{\text{norm}}(\hat{\mathbf{y}}^j \mid \mathbf{x}) \qquad (17)$$

We set $m = 10$. This is the same as the length-normalised predictive entropy baseline in Kuhn et al. (2023).

**Semantic Entropy.**  Semantic entropy is an entropy-based uncertainty measure which uses a bidirectional entailment algorithm for marginalising over semantically-equivalent samples (Kuhn et al., 2023). We follow the settings in Kuhn et al. (2023). Specifically, we use the multinomial sampling with a temperature of $0.5$ to obtain an answer list of size 10 for each input sequence for uncertainty

estimation. We use the Deberta-large model (He et al., 2020) that is fine-tuned on the NLI data set, MNLI (Williams et al., 2017) for the bidirectional entailment clustering algorithm.

**Self-eval.**  Self-eval is a simple baseline that obtains a selection score from the LLM by asking whether the proposed answer $\hat{\mathbf{y}}^*$ is correct or not. Suppose $\mathbf{z}_s$ is a series of tokens representing the self-evaluation trigger string "The answer is ". Suppose $z_c$ and $z_w$ are the tokens that represent the words "correct" and "wrong" respectively. Recall that the logits of the model $f$ on $v$ given $\mathbf{x}$ is $\bar{f}(v \mid \mathbf{x})$. Then, the self-eval score is computed as:

$$P(z_c \mid \mathbf{x}, \hat{\mathbf{y}}^*) = \frac{\exp\left(\bar{f}(z_c \mid \mathbf{x}, \hat{\mathbf{y}}^*, \mathbf{z}_s)\right)}{\sum_{z \in \{z_c, z_w\}} \exp\left(\bar{f}(z \mid \mathbf{x}, \hat{\mathbf{y}}^*, \mathbf{z}_s)\right)} \qquad (18)$$

**P(True).**  P(True) proposed by Kadavath et al. (2022) is a way to estimate the probability that a model's generation is correct by "asking" the model if its answer is correct. It samples $m$ answers and constructs a new natural language question using these possible answers as context before asking whether the proposed answer $\hat{\mathbf{y}}^*$ is correct and measures the probability of the completion being True. We set $m = 4$ and use the multinomial sampling with a temperature of $1.0$ to sample the answers. The format of the prompt is:

```
Question: Who was the third president of
the United States?
Here are some brainstormed ideas:
James Monroe
Thomas Jefferson
John Adams
Benjamin Harrison
George Washington
Possible Answer: James Monroe
Is the possible answer: (A) True (B) False.
The possible answer is:
```

where the "brainstormed answers" are from the set of sampled answers and P(True) (i.e., the likelihood of the next token being True) is taken as the uncertainty measure.

## E  Training Details

We have two stage training: the first stage is to train the soft prompt $\theta_p$ using the training objective (12) and the second stage is to train the soft prompt $\theta_s$ using the training objective (13). For

both stages, we train the soft prompt for 10 epochs using AdamW optimizer with a batch size of 8, a learning rate of 0.01 and cosine learning rate schedule. We remove those data points $(\mathbf{x}, \mathbf{y})$ where $|\mathbf{x}| + |\mathbf{y}| > 700$ from the training set $\mathcal{D}^{tr}$ to reduce GPU memory usage during training. Here, $|\mathbf{x}|$ is the length of the sequence $\mathbf{x}$. This only removes a very small portion of data points from the training set for each dataset (remove 4.02% training data points in CoQA, 0% training data points in TriviaQA and 0.04% training data points in SQuAD). During training $\theta_p$ or $\theta_s$, we always use 20% training data as validation data for selecting the best model among all checkpoints after each training epoch. Training $\theta_p$, we select the best model based on the loss on the validation data. When training $\theta_s$, we select the best model based on the AUROC on the validation data.

## F    Computational Complexity Analysis

The proposed method ASPIRE needs to train two soft prompts $\theta_p$ and $\theta_s$. The complexity of training $\theta_p$ using the training objective (12) is the same as the complexity of the standard soft prompt tuning. When training $\theta_s$ using the training objective (13), the number of training steps is the same as that of training $\theta_p$. In each training step of training $\theta_s$, we compute gradients for one correct output and two wrong outputs while in each training step of training $\theta_p$, we compute gradients for one reference output. Thus, the complexity of training $\theta_s$ is the same as that of training $\theta_p$. Therefore, the complexity of the proposed method ASPIRE in the training time is the same as that of the standard soft prompt tuning.

We analyze the computational complexity of different methods at test time in terms of the number of forward passes for the LLM. Since the LLM generates the output sequence in an auto-regressive way, the number of forward passes needed depends on the length of the generated output sequence. Suppose the maximum length of the generated output sequence is $l_{max}$. To generate an output sequence given an input sequence, we need one forward pass to encode the input sequence and at most $l_{max}$ forward passes to obtain the output sequence. Thus, for generating the output sequence, the maximum number of forward passes is $1 + l_{max}$ and the complexity is $O(l_{max})$. For the perplexity method, the computational complexity is $O(l_{max})$ since we only need additional one forward pass to obtain the

perplexity score. For the predictive entropy method, the computational complexity is $O(m \cdot l_{max})$ since we need to additionally generate $m$ answers and compute the likelihood of those $m$ answers. For the semantic entropy method, we omit the computational complexity of the bidirectional entailment clustering algorithm since its computational cost is much smaller than that of the generation of the LLM as stated in Kuhn et al. (2023). Thus, the computational complexity for semantic entropy is $O(m \cdot l_{max})$. For the self-eval method, the computational complexity is $O(l_{max})$ since we only need one additional forward pass to obtain the self-eval score. For the P(True) method, the computational complexity is $O(m \cdot l_{max})$ since we need to additionally generate $m$ answers and need one forward pass to compute the P(True) score. For the proposed method ASPIRE, the computational complexity is $O(l_{max})$ since we only need additional one forward pass to obtain the learned self-eval score. Table 6 summarizes the computational complexity of different methods at test time.

| Method | Complexity |
|---|---|
| Perplexity | $O(l_{max})$ |
| Predictive Entropy | $O(m \cdot l_{max})$ |
| Semantic Entropy | $O(m \cdot l_{max})$ |
| Self-eval | $O(l_{max})$ |
| P(True) | $O(m \cdot l_{max})$ |
| ASPIRE (ours) | $O(l_{max})$ |

Table 6: Computational complexity of different methods in the test time.

## G    Rouge Threshold for Evaluation

We use the Rouge-L (Lin and Och, 2004) metric to evaluate if the predicted answer is correct or not. The Rouge-L metric produces a score in $[0, 1]$. We need a threshold $\gamma$ to determine whether the predicted answer is correct or not. If the Rouge-L score is larger than the threshold $\gamma$, then the predicted answer is correct; otherwise, the predicted answer is wrong. The choice of $\gamma$ depends on the applications. Low values of $\gamma$ may lead to labeling some wrong answers as correct answers while large values of $\gamma$ may lead to labeling some correct answers as wrong answers. If we regard the wrong answer as the positive class, then we can use the precision and recall metrics to evaluate the choice of $\gamma$. To compute the precision and recall metrics, we need ground-truth labels for determining the

correctness of predicted answers, which requires manual labeling. If the Rouge-L score is equal to 0 (or 1), then it is mostly sure that the predicted answer is wrong (or correct). Thus, we only need to label those samples whose Rouge-L scores are in $(0, 1)$. To compare different values of $\gamma$, we compute the precision and recall metrics after manually label 200 samples whose Rouge-L scores are in the range of $(0, 1)$. The results in Table 7 show that larger values of $\gamma$ lead to higher recall but lower precision, while the lower values of $\gamma$ lead to higher precision but lower recall. We propose this work for safety-critical applications where accepting a wrong answer is more costly compared to rejecting a correct answer that is different from the reference answers. Thus, we prefer high recall than high precision. In our experiments, we evaluate different methods under the Rouge-L metric with $\gamma \in \{0.7, 0.8, 0, 9\}$ to ensure that the recall is at least 90%.

## H Rouge Threshold for the Proposed Framework

In the proposed framework ASPIRE, we need the Rouge threshold $\hat{\gamma}$ to determine if the generated answer is correct or not. We want to set a large enough value of $\hat{\gamma}$ so that the generated wrong answers won't be labeled as correct answers. To determine the value of $\hat{\gamma}$, we manually label the correctness of the 10 generated answers for 50 training examples from each dataset (we have three datasets CoQA, TriviaQA and SQuAD). The answers are generated using the OPT-2.7B model. We find that if we set $\hat{\gamma} = 0.9$, then no wrong answers would be labeled as correct answers. Thus, we set $\hat{\gamma} = 0.9$ for the proposed framework.

## I Complete Results

In this section, we present the complete results for OPT and GPT2 models of different sizes and different Rouge threshold $\gamma$. We first evaluate the accuracy of different LLMs. The results are in Table 8 (set $\gamma = 0.7$), Table 9 (set $\gamma = 0.8$) and Table 10 (set $\gamma = 0.9$). The results show that after training $\theta_p$ via soft prompt tuning, the accuracy of LLMs is improved significantly. We then evaluate different approaches to compute the selection score when the model's predictions are fixed. The results are in Table 11 (use GPT2 models and set $\gamma = 0.7$), Table 12 (use GPT2 models and set $\gamma = 0.8$), Table 13 (use GPT2 models and set $\gamma = 0.9$), Ta-

ble 14 (use OPT models and set $\gamma = 0.7$), Table 15 (use OPT models and set $\gamma = 0.8$) and Table 16 (use OPT models and set $\gamma = 0.9$). The results show that the proposed method ASPIRE significantly outperforms the baselines in terms of AUACC and AUROC across different datasets and LLMs for different values of the Rouge threshold $\gamma$.

## J The Effect of the Hyper-parameter $\alpha$

We study the effect of the hyper-parameter $\alpha$ in the proposed selection score (Eq. (11)) for our method. The results in Table 17 show that setting $\alpha = 0.25$ leads to the best performance across different datasets and different models. Only using the normalized likelihood (i.e., $\alpha = 0$) or only using the learned self-eval score (i.e., $\alpha = 1$) consistently leads to much worse performance. We choose $\alpha$ for our method based on the performance on the validation data from the TriviaQA dataset using the OPT-2.7B model. We then use the same $\alpha$ value for different datasets and different models. We consider $\alpha \in \{0.0, 0.25, 0.5, 0.75, 1.0\}$ when tuning it. Based on the validation results, we set $\alpha = 0.25$ by default.

## K Comparing with Self-Consistency

Self-consistency (Wang et al., 2022) can be used to obtain confidence measures as proposed by Si et al. (2022). We sample 10 times to obtain a set of different answers for each question using the multinomial sampling with a temperature of 0.5. Among all the generated answers, we take the most frequent answer as the final prediction and its frequency as the selection score. Since self-consistency produces discrete selection scores (in the above setting, the number of possible selection scores is 10) and we use the composite trapezoidal rule to compute AUACC, it is easier for self-consistency to achieve high AUACC compared to those approaches that produce continuous selection scores. Note that the proposed method produce continuous selection scores. Thus, it might not be fair to compare the proposed method with self-consistency. However, even though self-consistency has more advantages in achieving high AUACC, the proposed method ASPIRE still significantly outperforms self-consistency as shown in Table 18. We also observe that Self-Consistency might lead to worse accuracy meaning that the LLM can be consistently wrong.

| $\gamma$ | CoQA | | TriviaQA | | SQuAD | |
|---|---|---|---|---|---|---|
| | Precision ↑ | Recall ↑ | Precision ↑ | Recall ↑ | Precision ↑ | Recall ↑ |
| 0.1 | 100.00 | 0.00 | 100.00 | 0.62 | 100.00 | 7.91 |
| 0.2 | 100.00 | 10.00 | 100.00 | 2.50 | 100.00 | 34.53 |
| 0.3 | 100.00 | 22.50 | 100.00 | 11.88 | 98.55 | 48.92 |
| 0.4 | 100.00 | 45.62 | 97.01 | 40.62 | 93.58 | 73.38 |
| 0.5 | 97.98 | 60.62 | 97.09 | 62.50 | 85.94 | 79.14 |
| 0.6 | 97.41 | 70.62 | 96.19 | 63.12 | 84.73 | 79.86 |
| 0.7 | 93.51 | 90.00 | 86.81 | 98.75 | 76.16 | 94.24 |
| 0.8 | 86.59 | 96.88 | 81.22 | 100.00 | 73.66 | 98.56 |
| 0.9 | 80.71 | 99.38 | 80.00 | 100.00 | 69.85 | 100.00 |

Table 7: Results of comparing different choices of the Rouge threshold $\gamma$. The wrong answer is regarded as the positive class. We use the OPT-2.7B model. We manually label 200 samples with Rouge-L scores in the range of (0, 1) in each dataset and then compute the precision and recall. All numbers are percentages.

| Model | CoQA | TriviaQA | SQuAD |
|---|---|---|---|
| | Acc ↑ | Acc ↑ | Acc ↑ |
| Pre-trained GPT2-Medium | 35.12 | 5.44 | 4.42 |
| Adapted GPT2-Medium with $\theta_p$ | 57.90 | 9.04 | 66.63 |
| Pre-trained GPT2-Large | 41.21 | 8.16 | 6.09 |
| Adapted GPT2-Large with $\theta_p$ | 63.89 | 12.50 | 71.34 |
| Pre-trained GPT2-XL | 46.27 | 11.80 | 7.41 |
| Adapted GPT2-XL with $\theta_p$ | 69.18 | 17.45 | 75.44 |
| Pre-trained OPT-350M | 28.76 | 4.35 | 13.65 |
| Adapted OPT-350M with $\theta_p$ | 59.46 | 8.25 | 64.74 |
| Pre-trained OPT-1.3B | 54.13 | 15.80 | 30.23 |
| Adapted OPT-1.3B with $\theta_p$ | 76.85 | 21.73 | 80.94 |
| Pre-trained OPT-2.7B | 60.68 | 21.38 | 35.95 |
| Adapted OPT-2.7B with $\theta_p$ | 80.45 | 29.21 | 83.27 |

Table 8: Results of evaluating the accuracy of different LLMs when the Rouge threshold $\gamma = 0.7$. All numbers are percentages.

## L  Qualitative Evaluation

We present some concrete examples from the TriviaQA dataset to show the advantages of the proposed method qualitatively. We compare the proposed method ASPIRE to the baseline Semantic Entropy. The model for generating answers is the adapted OPT-2.7B with learned $\theta_p$. The examples below show that some semantic entropy scores for correct predictions are lower than some semantic entropy scores for wrong predictions while the AS-PIRE scores for correct predictions are consistently higher than the ASPIRE scores for wrong predictions. The ASPIRE scores are log likelihood scores and can be converted to likelihood scores by taking exponentiation with the base $e$.

**Examples where predictions are correct**

Question: Who is the most successful UK solo artist in the USA?
Answer: Elton John.
Predicted answer: Elton John.
Semantic entropy score: -1.1031
ASPIRE score: -0.8163

Question: In which decade of the 20th century was Anne Bancroft born?
Answer: 1930s.
Predicted answer: 1930s.
Semantic entropy score: -0.6167
ASPIRE score: -0.9026

Question: The Suez Canal connects the Mediterranean Sea to which other Sea?
Answer: Red sea.

| Model | CoQA | TriviaQA | SQuAD |
|---|---|---|---|
| | Acc ↑ | Acc ↑ | Acc ↑ |
| Pre-trained GPT2-Medium | 32.12 | 5.00 | 2.85 |
| Adapted GPT2-Medium with $\theta_p$ | 55.12 | 8.71 | 62.92 |
| Pre-trained GPT2-Large | 38.16 | 7.64 | 3.98 |
| Adapted GPT2-Large with $\theta_p$ | 61.04 | 12.14 | 67.56 |
| Pre-trained GPT2-XL | 42.67 | 11.10 | 5.21 |
| Adapted GPT2-XL with $\theta_p$ | 66.49 | 16.96 | 71.17 |
| Pre-trained OPT-350M | 27.38 | 4.25 | 11.15 |
| Adapted OPT-350M with $\theta_p$ | 57.02 | 8.05 | 61.29 |
| Pre-trained OPT-1.3B | 51.35 | 15.35 | 25.73 |
| Adapted OPT-1.3B with $\theta_p$ | 74.46 | 21.26 | 77.28 |
| Pre-trained OPT-2.7B | 57.72 | 20.71 | 30.94 |
| Adapted OPT-2.7B with $\theta_p$ | 77.97 | 28.55 | 80.04 |

Table 9: Results of evaluating the accuracy of different LLMs when the Rouge threshold $\gamma = 0.8$. All numbers are percentages.

| Model | CoQA | TriviaQA | SQuAD |
|---|---|---|---|
| | Acc ↑ | Acc ↑ | Acc ↑ |
| Pre-trained GPT2-Medium | 30.49 | 4.88 | 1.99 |
| Adapted GPT2-Medium with $\theta_p$ | 53.11 | 8.53 | 60.51 |
| Pre-trained GPT2-Large | 36.20 | 7.41 | 3.00 |
| Adapted GPT2-Large with $\theta_p$ | 59.04 | 11.85 | 64.98 |
| Pre-trained GPT2-XL | 40.32 | 10.82 | 4.12 |
| Adapted GPT2-XL with $\theta_p$ | 64.59 | 16.70 | 68.83 |
| Pre-trained OPT-350M | 26.81 | 4.20 | 9.62 |
| Adapted OPT-350M with $\theta_p$ | 55.33 | 8.00 | 59.35 |
| Pre-trained OPT-1.3B | 49.78 | 15.24 | 22.79 |
| Adapted OPT-1.3B with $\theta_p$ | 72.78 | 21.07 | 74.97 |
| Pre-trained OPT-2.7B | 56.06 | 20.55 | 27.41 |
| Adapted OPT-2.7B with $\theta_p$ | 76.45 | 28.26 | 78.12 |

Table 10: Results of evaluating the accuracy of different LLMs when the Rouge threshold $\gamma = 0.9$. All numbers are percentages.

Predicted answer: Red Sea.
Semantic entropy score: 2.2082
ASPIRE score: -0.2309

Question: Sun Yat Sen overthrew the emperor in which country establishing a republic after 2000 years of imperial rule?
Answer: China.
Predicted answer: China.
Semantic entropy score: 2.0028
ASPIRE score: -0.4205

**Examples where predictions are wrong**
Question: Who was the director of the CIA from 1976-81?
Answer: George Bush.

Predicted answer: George H W Bush.
Semantic entropy score: 0.4547
ASPIRE score: -1.0397

Question: What Michelle Pfeiffer movie got a boost from the Coolio song Gangsta's Paradise?
Answer: Dangerous Minds.
Predicted answer: Scarface.
Semantic entropy score: 0.0647
ASPIRE score: -1.0531

Question: What was President Gerald Ford's middle name?
Answer: Rudolph.
Predicted answer: William.

| Model | Method | CoQA | | TriviaQA | | SQuAD | |
|---|---|---|---|---|---|---|---|
| | | AUACC ↑ | AUROC ↑ | AUACC ↑ | AUROC ↑ | AUACC ↑ | AUROC ↑ |
| Pre-trained GPT2-Medium | Perplexity | 38.92 | 55.77 | 7.67 | 60.52 | 4.58 | 55.35 |
| | Predictive Entropy | 45.91 | 62.89 | 10.02 | 67.66 | 5.99 | 57.07 |
| | Semantic Entropy | 48.35 | 66.30 | 10.28 | 68.54 | 6.18 | 57.36 |
| | Self-eval | 36.16 | 51.14 | 6.26 | 56.74 | 3.70 | 43.44 |
| | P(True) | 34.08 | 48.21 | 8.24 | 60.62 | 5.41 | 54.33 |
| Adapted GPT2-Medium with $\theta_p$ | Perplexity | 72.03 | 67.89 | 18.02 | 72.17 | 81.91 | 72.38 |
| | Predictive Entropy | 72.59 | 69.42 | 20.07 | 77.48 | 82.00 | 73.09 |
| | Semantic Entropy | 73.95 | 71.54 | 19.86 | 77.62 | 82.35 | 73.66 |
| | Self-eval | 57.94 | 50.43 | 9.94 | 54.68 | 64.79 | 46.99 |
| | P(True) | 56.71 | 48.76 | 13.55 | 60.79 | 65.94 | 49.13 |
| | ASPIRE (ours) | **76.32** | **75.30** | **20.65** | **79.41** | **84.15** | **77.59** |
| Pre-trained GPT2-Large | Perplexity | 48.57 | 59.82 | 13.74 | 66.51 | 6.39 | 53.96 |
| | Predictive Entropy | 55.04 | 66.68 | 16.25 | 70.46 | 8.25 | 57.03 |
| | Semantic Entropy | 57.13 | 69.57 | 16.02 | 70.06 | 8.81 | 59.24 |
| | Self-eval | 42.24 | 51.72 | 9.78 | 54.74 | 5.07 | 46.79 |
| | P(True) | 36.73 | 45.69 | 8.60 | 48.62 | 6.83 | 55.62 |
| Adapted GPT2-Large with $\theta_p$ | Perplexity | 77.15 | 68.15 | 26.83 | 77.06 | 86.26 | 75.34 |
| | Predictive Entropy | 77.45 | 69.76 | 27.83 | 80.02 | 86.32 | 75.65 |
| | Semantic Entropy | 78.85 | 71.97 | 27.61 | 79.88 | 86.53 | 75.90 |
| | Self-eval | 64.28 | 50.61 | 14.26 | 54.34 | 70.86 | 50.81 |
| | P(True) | 58.97 | 45.55 | 12.38 | 47.61 | 70.73 | 50.09 |
| | ASPIRE (ours) | **81.30** | **76.38** | **29.13** | **82.14** | **87.83** | **79.22** |
| Pre-trained GPT2-XL | Perplexity | 55.93 | 62.05 | 22.60 | 72.88 | 7.68 | 51.90 |
| | Predictive Entropy | 60.76 | 67.53 | 24.83 | 76.20 | 10.04 | 57.21 |
| | Semantic Entropy | 63.03 | 70.50 | 24.37 | 75.33 | 10.38 | 59.17 |
| | Self-eval | 46.67 | 50.83 | 9.30 | 42.75 | 7.32 | 49.56 |
| | P(True) | 46.98 | 51.17 | 10.62 | 44.54 | 10.69 | 60.87 |
| Adapted GPT2-XL with $\theta_p$ | Perplexity | 83.27 | 72.79 | 36.49 | 79.92 | 88.73 | 75.08 |
| | Predictive Entropy | 83.49 | 73.44 | 37.31 | 82.21 | 88.25 | 74.16 |
| | Semantic Entropy | 84.40 | 75.16 | 36.68 | 81.40 | 88.62 | 75.26 |
| | Self-eval | 69.91 | 51.90 | 14.39 | 43.33 | 74.26 | 49.13 |
| | P(True) | 70.63 | 52.83 | 13.59 | 40.59 | 74.34 | 49.09 |
| | ASPIRE (ours) | **85.65** | **78.32** | **38.06** | **83.23** | **89.86** | **78.35** |

Table 11: Results of evaluating different methods to compute the selection score when the model's predictions are fixed. We use the GPT2 models and set the Rouge threshold $\gamma = 0.7$. All numbers are percentages. **Bold** numbers are superior results.

Semantic entropy score: -3.9773
ASPIRE score: -2.8203

Question: Kim Carnes' nine weeks at No 1 with Bette Davis Eyes was interrupted for one week by which song?
Answer: Stars on 45 medley.
Predicted answer: Bette Davis Eyes.
Semantic entropy score: -1.4973
ASPIRE score: -2.2803

| Model | Method | CoQA | | TriviaQA | | SQuAD | |
|---|---|---|---|---|---|---|---|
| | | AUACC ↑ | AUROC ↑ | AUACC ↑ | AUROC ↑ | AUACC ↑ | AUROC ↑ |
| Pre-trained GPT2-Medium | Perplexity | 35.24 | 54.28 | 7.03 | 59.54 | 2.82 | 53.29 |
| | Predictive Entropy | 42.43 | 62.42 | 9.23 | 66.62 | 3.86 | 58.15 |
| | Semantic Entropy | 45.53 | 66.84 | 9.52 | 67.83 | 4.02 | 58.53 |
| | Self-eval | 32.97 | 51.13 | 5.95 | 57.98 | 2.06 | 40.62 |
| | P(True) | 31.05 | 48.10 | 7.81 | 61.51 | 3.73 | 55.72 |
| Adapted GPT2-Medium with $\theta_p$ | Perplexity | 69.36 | 67.21 | 17.42 | 71.77 | 79.26 | 72.25 |
| | Predictive Entropy | 69.74 | 68.58 | 19.38 | 77.26 | 79.18 | 72.70 |
| | Semantic Entropy | 71.35 | 71.10 | 19.22 | 77.55 | 79.61 | 73.53 |
| | Self-eval | 55.04 | 50.31 | 9.75 | 55.26 | 61.43 | 47.61 |
| | P(True) | 53.95 | 48.66 | 13.32 | 61.55 | 62.07 | 49.06 |
| | ASPIRE (ours) | **73.97** | **75.05** | **20.12** | **79.59** | **82.02** | **78.02** |
| Pre-trained GPT2-Large | Perplexity | 44.95 | 58.70 | 13.06 | 66.47 | 4.06 | 51.95 |
| | Predictive Entropy | 51.57 | 66.32 | 15.43 | 70.33 | 5.61 | 57.34 |
| | Semantic Entropy | 54.39 | 70.24 | 15.25 | 70.08 | 6.25 | 61.09 |
| | Self-eval | 39.66 | 52.36 | 9.21 | 54.62 | 3.15 | 45.40 |
| | P(True) | 33.73 | 45.72 | 8.20 | 49.18 | 4.68 | 57.51 |
| Adapted GPT2-Large with $\theta_p$ | Perplexity | 74.64 | 67.40 | 26.20 | 76.89 | 83.61 | 74.57 |
| | Predictive Entropy | 74.96 | 69.07 | 27.22 | 80.01 | 83.57 | 74.67 |
| | Semantic Entropy | 76.65 | 71.82 | 27.06 | 79.99 | 83.81 | 75.10 |
| | Self-eval | 61.88 | 50.99 | 13.83 | 54.15 | 67.28 | 51.20 |
| | P(True) | 56.35 | 45.90 | 11.95 | 47.55 | 67.13 | 50.52 |
| | ASPIRE (ours) | **79.39** | **76.49** | **28.43** | **82.01** | **85.71** | **79.27** |
| Pre-trained GPT2-XL | Perplexity | 52.07 | 61.15 | 21.54 | 72.72 | 5.30 | 49.81 |
| | Predictive Entropy | 56.83 | 66.90 | 23.65 | 76.15 | 7.27 | 56.53 |
| | Semantic Entropy | 59.74 | 70.83 | 23.23 | 75.38 | 7.59 | 58.85 |
| | Self-eval | 43.34 | 51.14 | 8.81 | 42.76 | 5.45 | 51.47 |
| | P(True) | 43.24 | 51.09 | 9.81 | 43.94 | 8.54 | 65.61 |
| Adapted GPT2-XL with $\theta_p$ | Perplexity | 81.05 | 71.85 | 35.61 | 79.69 | 86.08 | 74.71 |
| | Predictive Entropy | 81.23 | 72.42 | 36.42 | 82.01 | 85.53 | 73.62 |
| | Semantic Entropy | 82.38 | 74.62 | 35.84 | 81.31 | 85.93 | 74.84 |
| | Self-eval | 67.35 | 51.89 | 14.05 | 43.45 | 70.30 | 49.21 |
| | P(True) | 68.02 | 52.83 | 13.21 | 40.48 | 69.47 | 48.32 |
| | ASPIRE (ours) | **83.91** | **78.09** | **37.26** | **83.18** | **87.76** | **78.82** |

Table 12: Results of evaluating different methods to compute the selection score when the model's predictions are fixed. We use the GPT2 models and set the Rouge threshold $\gamma = 0.8$. All numbers are percentages. **Bold** numbers are superior results.

| Model | Method | CoQA | | TriviaQA | | SQuAD | |
|---|---|---|---|---|---|---|---|
| | | AUACC ↑ | AUROC ↑ | AUACC ↑ | AUROC ↑ | AUACC ↑ | AUROC ↑ |
| Pre-trained GPT2-Medium | Perplexity | 33.09 | 53.03 | 6.72 | 58.75 | 1.74 | 47.52 |
| | Predictive Entropy | 40.52 | 62.05 | 8.90 | 66.11 | 2.61 | 56.43 |
| | Semantic Entropy | 44.11 | 67.38 | 9.26 | 67.57 | 2.85 | 57.36 |
| | Self-eval | 31.46 | 51.29 | 5.88 | 58.50 | 1.17 | 35.24 |
| | P(True) | 29.24 | 47.97 | 7.73 | 62.30 | 2.78 | 58.35 |
| Adapted GPT2-Medium with $\theta_p$ | Perplexity | 67.42 | 66.51 | 17.06 | 71.50 | 77.63 | 72.22 |
| | Predictive Entropy | 67.89 | 68.15 | 19.06 | 77.30 | 77.49 | 72.55 |
| | Semantic Entropy | 69.77 | 71.20 | 18.94 | 77.75 | 78.07 | 73.77 |
| | Self-eval | 53.15 | 50.51 | 9.67 | 55.80 | 58.95 | 47.61 |
| | P(True) | 51.95 | 48.59 | 13.21 | 62.06 | 59.55 | 48.95 |
| | ASPIRE (ours) | **72.39** | **74.96** | **19.81** | **79.64** | **80.68** | **78.49** |
| Pre-trained GPT2-Large | Perplexity | 42.74 | 57.93 | 12.56 | 65.85 | 2.78 | 46.87 |
| | Predictive Entropy | 49.68 | 66.44 | 14.89 | 69.76 | 3.94 | 54.80 |
| | Semantic Entropy | 52.90 | 71.07 | 14.76 | 69.63 | 4.53 | 59.75 |
| | Self-eval | 38.08 | 52.84 | 8.97 | 54.45 | 2.42 | 45.97 |
| | P(True) | 31.71 | 45.48 | 8.06 | 49.66 | 3.84 | 60.48 |
| Adapted GPT2-Large with $\theta_p$ | Perplexity | 72.97 | 66.96 | 25.67 | 76.60 | 81.77 | 74.01 |
| | Predictive Entropy | 73.34 | 68.78 | 26.69 | 79.84 | 81.80 | 74.31 |
| | Semantic Entropy | 75.24 | 71.99 | 26.59 | 79.96 | 82.29 | 75.36 |
| | Self-eval | 60.35 | 51.58 | 13.44 | 53.82 | 64.89 | 51.42 |
| | P(True) | 54.34 | 45.68 | 11.65 | 47.57 | 64.77 | 50.75 |
| | ASPIRE (ours) | **78.08** | **76.61** | **27.97** | **81.99** | **84.24** | **79.37** |
| Pre-trained GPT2-XL | Perplexity | 49.71 | 60.59 | 20.96 | 72.31 | 4.04 | 46.28 |
| | Predictive Entropy | 54.74 | 67.05 | 23.10 | 75.93 | 5.78 | 55.59 |
| | Semantic Entropy | 58.07 | 71.83 | 22.76 | 75.33 | 6.18 | 59.20 |
| | Self-eval | 41.19 | 51.46 | 8.67 | 42.97 | 4.61 | 53.48 |
| | P(True) | 40.37 | 50.77 | 9.46 | 43.58 | 7.30 | 69.47 |
| Adapted GPT2-XL with $\theta_p$ | Perplexity | 79.60 | 71.40 | 35.11 | 79.47 | 84.69 | 74.62 |
| | Predictive Entropy | 79.86 | 72.25 | 35.93 | 81.85 | 84.23 | 73.81 |
| | Semantic Entropy | 81.25 | 74.87 | 35.39 | 81.19 | 84.74 | 75.38 |
| | Self-eval | 65.61 | 52.08 | 13.87 | 43.51 | 68.10 | 49.56 |
| | P(True) | 65.90 | 52.61 | 12.95 | 40.31 | 67.37 | 48.74 |
| | ASPIRE (ours) | **82.77** | **78.11** | **36.81** | **83.09** | **86.57** | **79.06** |

Table 13: Results of evaluating different methods to compute the selection score when the model's predictions are fixed. We use the GPT2 models and set the Rouge threshold $\gamma = 0.9$. All numbers are percentages. **Bold** numbers are superior results.

| Model | Method | CoQA | | TriviaQA | | SQuAD | |
|---|---|---|---|---|---|---|---|
| | | AUACC ↑ | AUROC ↑ | AUACC ↑ | AUROC ↑ | AUACC ↑ | AUROC ↑ |
| Pre-trained OPT-350M | Perplexity | 35.37 | 59.39 | 6.81 | 67.09 | 13.07 | 50.34 |
| | Predictive Entropy | 36.55 | 60.31 | 7.20 | 65.04 | 17.86 | 59.33 |
| | Semantic Entropy | 38.80 | 64.38 | 7.31 | 65.15 | 19.08 | 61.66 |
| | Self-eval | 30.02 | 52.69 | 5.98 | 61.17 | 14.00 | 51.41 |
| | P(True) | 28.70 | 50.60 | 5.29 | 55.69 | 17.76 | 59.55 |
| Adapted OPT-350M with $\theta_p$ | Perplexity | 74.50 | 70.21 | 18.13 | 75.86 | 80.64 | 73.76 |
| | Predictive Entropy | 74.14 | 68.88 | 18.73 | 76.83 | 80.79 | 73.46 |
| | Semantic Entropy | 74.94 | 70.14 | 18.46 | 76.91 | 81.10 | 73.98 |
| | Self-eval | 60.86 | 51.67 | 10.29 | 57.89 | 65.48 | 50.70 |
| | P(True) | 59.20 | 50.04 | 8.71 | 52.05 | 64.55 | 50.29 |
| | ASPIRE (ours) | **75.55** | **72.37** | **19.00** | **78.54** | **82.59** | **77.18** |
| Pre-trained OPT-1.3B | Perplexity | 69.51 | 69.32 | 29.78 | 74.77 | 32.43 | 54.65 |
| | Predictive Entropy | 69.46 | 68.48 | 31.01 | 75.21 | 41.06 | 62.96 |
| | Semantic Entropy | 70.42 | 70.46 | 30.63 | 74.74 | 43.33 | 66.30 |
| | Self-eval | 56.38 | 52.86 | 15.06 | 49.96 | 30.74 | 51.50 |
| | P(True) | 57.21 | 53.19 | 16.83 | 51.19 | 28.88 | 46.75 |
| Adapted OPT-1.3B with $\theta_p$ | Perplexity | 88.50 | 73.64 | 42.46 | 79.96 | 91.45 | 74.47 |
| | Predictive Entropy | 88.24 | 72.38 | 43.03 | 80.46 | 91.46 | 74.38 |
| | Semantic Entropy | 88.91 | 74.02 | 42.70 | 80.02 | 91.72 | 75.44 |
| | Self-eval | 78.52 | 53.08 | 20.65 | 49.24 | 81.05 | 51.52 |
| | P(True) | 79.07 | 52.76 | 22.20 | 50.34 | 81.58 | 50.77 |
| | ASPIRE (ours) | **90.76** | **79.26** | **44.03** | **83.06** | **93.41** | **81.17** |
| Pre-trained OPT-2.7B | Perplexity | 75.26 | 70.16 | 40.93 | 78.86 | 40.82 | 57.20 |
| | Predictive Entropy | 75.29 | 69.16 | 41.20 | 78.92 | 47.18 | 62.85 |
| | Semantic Entropy | 76.31 | 70.96 | 40.72 | 78.06 | 51.53 | 68.40 |
| | Self-eval | 62.32 | 52.26 | 25.88 | 59.04 | 41.78 | 59.05 |
| | P(True) | 62.16 | 51.80 | 24.88 | 56.89 | 34.77 | 49.42 |
| Adapted OPT-2.7B with $\theta_p$ | Perplexity | 90.80 | 74.23 | 53.56 | 81.74 | 92.86 | 75.72 |
| | Predictive Entropy | 90.63 | 72.87 | 53.91 | 82.19 | 92.96 | 75.58 |
| | Semantic Entropy | 91.23 | 74.61 | 53.58 | 81.55 | 93.21 | 76.53 |
| | Self-eval | 81.30 | 50.76 | 32.98 | 56.03 | 86.34 | 56.99 |
| | P(True) | 81.14 | 51.01 | 33.48 | 56.27 | 82.59 | 49.48 |
| | ASPIRE (ours) | **92.63** | **80.25** | **55.06** | **84.44** | **94.73** | **82.60** |

Table 14: Results of evaluating different methods to compute the selection score when the model's predictions are fixed. We use the OPT models and set the Rouge threshold $\gamma = 0.7$. All numbers are percentages. **Bold** numbers are superior results.

| Model | Method | CoQA | | TriviaQA | | SQuAD | |
|---|---|---|---|---|---|---|---|
| | | AUACC ↑ | AUROC ↑ | AUACC ↑ | AUROC ↑ | AUACC ↑ | AUROC ↑ |
| Pre-trained OPT-350M | Perplexity | 33.50 | 58.50 | 6.64 | 66.73 | 10.50 | 49.05 |
| | Predictive Entropy | 34.88 | 59.82 | 7.03 | 64.84 | 14.59 | 58.98 |
| | Semantic Entropy | 37.51 | 64.67 | 7.17 | 65.20 | 15.74 | 61.85 |
| | Self-eval | 28.73 | 52.91 | 5.93 | 61.83 | 11.33 | 50.71 |
| | P(True) | 27.10 | 50.36 | 5.25 | 56.28 | 15.38 | 61.06 |
| Adapted OPT-350M with $\theta_p$ | Perplexity | 72.04 | 69.26 | 17.76 | 75.70 | 77.72 | 72.95 |
| | Predictive Entropy | 71.77 | 68.12 | 18.30 | 76.51 | 77.91 | 72.76 |
| | Semantic Entropy | 72.80 | 69.90 | 18.06 | 76.67 | 78.35 | 73.57 |
| | Self-eval | 58.65 | 52.06 | 10.03 | 57.87 | 61.83 | 50.32 |
| | P(True) | 56.82 | 50.17 | 8.58 | 52.27 | 61.13 | 50.18 |
| | ASPIRE (ours) | **73.56** | **72.05** | **18.63** | **78.42** | **80.33** | **77.29** |
| Pre-trained OPT-1.3B | Perplexity | 66.09 | 67.76 | 29.01 | 74.46 | 27.67 | 53.61 |
| | Predictive Entropy | 66.34 | 67.36 | 30.21 | 74.92 | 35.65 | 62.44 |
| | Semantic Entropy | 67.64 | 70.02 | 29.91 | 74.61 | 38.00 | 66.50 |
| | Self-eval | 53.87 | 53.23 | 14.73 | 50.12 | 26.42 | 51.63 |
| | P(True) | 54.07 | 52.70 | 16.44 | 51.38 | 23.69 | 45.44 |
| Adapted OPT-1.3B with $\theta_p$ | Perplexity | 86.67 | 72.53 | 41.59 | 79.61 | 89.00 | 73.48 |
| | Predictive Entropy | 86.41 | 71.33 | 42.18 | 80.15 | 89.02 | 73.35 |
| | Semantic Entropy | 87.27 | 73.41 | 41.89 | 79.75 | 89.42 | 74.81 |
| | Self-eval | 76.49 | 53.45 | 20.23 | 49.20 | 77.85 | 52.25 |
| | P(True) | 76.79 | 52.52 | 21.65 | 50.25 | 77.86 | 50.71 |
| | ASPIRE (ours) | **89.48** | **79.05** | **43.23** | **82.84** | **91.86** | **81.44** |
| Pre-trained OPT-2.7B | Perplexity | 72.00 | 68.49 | 39.79 | 78.43 | 35.76 | 56.78 |
| | Predictive Entropy | 72.23 | 67.89 | 40.05 | 78.49 | 41.18 | 61.98 |
| | Semantic Entropy | 73.64 | 70.43 | 39.67 | 77.81 | 45.83 | 68.35 |
| | Self-eval | 59.51 | 52.24 | 25.10 | 59.02 | 36.71 | 59.36 |
| | P(True) | 58.81 | 51.26 | 24.13 | 56.80 | 29.13 | 48.41 |
| Adapted OPT-2.7B with $\theta_p$ | Perplexity | 89.10 | 73.16 | 52.64 | 81.56 | 91.04 | 74.96 |
| | Predictive Entropy | 88.95 | 72.00 | 52.97 | 82.00 | 91.16 | 74.86 |
| | Semantic Entropy | 89.80 | 74.53 | 52.71 | 81.47 | 91.46 | 75.91 |
| | Self-eval | 79.12 | 51.00 | 32.28 | 56.03 | 83.28 | 56.52 |
| | P(True) | 78.74 | 50.89 | 32.95 | 56.42 | 79.05 | 49.26 |
| | ASPIRE (ours) | **91.49** | **80.12** | **54.15** | **84.28** | **93.37** | **82.33** |

Table 15: Results of evaluating different methods to compute the selection score when the model's predictions are fixed. We use the OPT models and set the Rouge threshold $\gamma = 0.8$. All numbers are percentages. **Bold** numbers are superior results.

| Model | Method | CoQA | | TriviaQA | | SQuAD | |
|---|---|---|---|---|---|---|---|
| | | AUACC ↑ | AUROC ↑ | AUACC ↑ | AUROC ↑ | AUACC ↑ | AUROC ↑ |
| Pre-trained OPT-350M | Perplexity | 32.58 | 57.88 | 6.50 | 66.50 | 8.63 | 46.42 |
| | Predictive Entropy | 34.13 | 59.61 | 6.91 | 64.62 | 12.56 | 58.33 |
| | Semantic Entropy | 36.97 | 64.81 | 7.06 | 65.06 | 13.82 | 61.99 |
| | Self-eval | 27.98 | 52.97 | 5.90 | 62.10 | 10.08 | 51.66 |
| | P(True) | 26.41 | 50.23 | 5.21 | 56.37 | 14.01 | 62.76 |
| Adapted OPT-350M with $\theta_p$ | Perplexity | 70.07 | 68.20 | 17.67 | 75.63 | 76.12 | 72.40 |
| | Predictive Entropy | 69.97 | 67.41 | 18.22 | 76.47 | 76.44 | 72.48 |
| | Semantic Entropy | 71.38 | 69.93 | 17.98 | 76.65 | 77.11 | 73.81 |
| | Self-eval | 57.05 | 52.30 | 9.96 | 57.81 | 59.96 | 50.56 |
| | P(True) | 55.10 | 50.32 | 8.53 | 52.22 | 59.05 | 50.03 |
| | ASPIRE (ours) | **71.94** | **71.41** | **18.54** | **78.39** | **79.12** | **77.38** |
| Pre-trained OPT-1.3B | Perplexity | 64.03 | 66.70 | 28.77 | 74.31 | 24.05 | 51.41 |
| | Predictive Entropy | 64.59 | 66.80 | 29.98 | 74.81 | 31.35 | 60.95 |
| | Semantic Entropy | 66.29 | 70.03 | 29.72 | 74.56 | 34.05 | 66.05 |
| | Self-eval | 52.35 | 53.37 | 14.64 | 50.14 | 24.12 | 52.63 |
| | P(True) | 52.51 | 52.64 | 16.27 | 51.38 | 20.92 | 45.41 |
| Adapted OPT-1.3B with $\theta_p$ | Perplexity | 85.21 | 71.30 | 41.21 | 79.43 | 87.71 | 73.17 |
| | Predictive Entropy | 85.05 | 70.44 | 41.81 | 80.00 | 87.81 | 73.34 |
| | Semantic Entropy | 86.23 | 73.38 | 41.55 | 79.66 | 88.24 | 74.81 |
| | Self-eval | 75.09 | 53.72 | 20.07 | 49.23 | 75.80 | 52.60 |
| | P(True) | 75.16 | 52.38 | 21.44 | 50.22 | 75.83 | 51.10 |
| | ASPIRE (ours) | **88.49** | **78.52** | **42.88** | **82.70** | **90.79** | **81.34** |
| Pre-trained OPT-2.7B | Perplexity | 70.07 | 67.37 | 39.42 | 78.23 | 31.18 | 54.43 |
| | Predictive Entropy | 70.44 | 67.03 | 39.69 | 78.34 | 36.14 | 60.36 |
| | Semantic Entropy | 72.29 | 70.35 | 39.34 | 77.68 | 40.96 | 67.71 |
| | Self-eval | 57.76 | 52.07 | 24.85 | 58.93 | 32.56 | 59.52 |
| | P(True) | 57.06 | 50.98 | 23.96 | 56.74 | 25.64 | 48.02 |
| Adapted OPT-2.7B with $\theta_p$ | Perplexity | 88.06 | 72.44 | 52.12 | 81.33 | 90.01 | 74.59 |
| | Predictive Entropy | 87.95 | 71.48 | 52.48 | 81.81 | 90.17 | 74.71 |
| | Semantic Entropy | 88.96 | 74.50 | 52.28 | 81.35 | 90.47 | 75.75 |
| | Self-eval | 77.71 | 51.04 | 31.90 | 55.89 | 81.27 | 56.36 |
| | P(True) | 77.16 | 50.54 | 32.62 | 56.33 | 76.89 | 48.85 |
| | ASPIRE (ours) | **90.76** | **79.94** | **53.68** | **84.10** | **92.52** | **82.04** |

Table 16: Results of evaluating different methods to compute the selection score when the model's predictions are fixed. We use the OPT models and set the Rouge threshold $\gamma = 0.9$. All numbers are percentages. **Bold** numbers are superior results.

| Model | Method | CoQA | | TriviaQA | | SQuAD | |
|---|---|---|---|---|---|---|---|
| | | AUACC ↑ | AUROC ↑ | AUACC ↑ | AUROC ↑ | AUACC ↑ | AUROC ↑ |
| Adapted GPT2-Medium with $\theta_p$ | ASPIRE ($\alpha = 0.0$) | 72.03 | 67.89 | 18.02 | 72.17 | 81.91 | 72.38 |
| | ASPIRE ($\alpha = 0.25$) | **76.32** | **75.30** | **20.65** | 79.41 | **84.15** | **77.59** |
| | ASPIRE ($\alpha = 0.5$) | 75.76 | 75.27 | 20.24 | **80.35** | 83.24 | 76.50 |
| | ASPIRE ($\alpha = 0.75$) | 73.26 | 71.99 | 18.01 | 77.00 | 81.76 | 73.98 |
| | ASPIRE ($\alpha = 1.0$) | 67.61 | 66.72 | 14.52 | 72.52 | 79.60 | 70.52 |
| Adapted GPT2-Large with $\theta_p$ | ASPIRE ($\alpha = 0.0$) | 77.15 | 68.15 | 26.83 | 77.06 | 86.26 | 75.34 |
| | ASPIRE ($\alpha = 0.25$) | **81.30** | 76.38 | **29.13** | 82.14 | **87.83** | **79.22** |
| | ASPIRE ($\alpha = 0.5$) | 80.87 | **76.39** | 28.49 | **82.41** | 86.97 | 77.66 |
| | ASPIRE ($\alpha = 0.75$) | 78.91 | 73.38 | 25.32 | 78.74 | 85.66 | 74.95 |
| | ASPIRE ($\alpha = 1.0$) | 74.22 | 68.01 | 19.77 | 72.75 | 83.99 | 71.74 |
| Adapted GPT2-XL with $\theta_p$ | ASPIRE ($\alpha = 0.0$) | 83.27 | 72.79 | 36.49 | 79.92 | 88.73 | 75.08 |
| | ASPIRE ($\alpha = 0.25$) | **85.65** | **78.32** | **38.06** | **83.23** | **89.86** | **78.35** |
| | ASPIRE ($\alpha = 0.5$) | 85.15 | 78.02 | 37.22 | 82.80 | 88.82 | 76.12 |
| | ASPIRE ($\alpha = 0.75$) | 83.03 | 74.22 | 33.37 | 78.17 | 87.47 | 73.13 |
| | ASPIRE ($\alpha = 1.0$) | 77.66 | 66.38 | 25.49 | 70.09 | 85.88 | 69.89 |
| Adapted OPT-350M with $\theta_p$ | ASPIRE ($\alpha = 0.0$) | 74.50 | 70.21 | 18.13 | 75.86 | 80.64 | 73.76 |
| | ASPIRE ($\alpha = 0.25$) | **75.55** | **72.37** | **19.00** | 78.54 | **82.59** | **77.18** |
| | ASPIRE ($\alpha = 0.5$) | 74.95 | 72.07 | 18.81 | **80.48** | 81.69 | 75.90 |
| | ASPIRE ($\alpha = 0.75$) | 72.55 | 68.45 | 16.21 | 78.76 | 80.16 | 73.27 |
| | ASPIRE ($\alpha = 1.0$) | 68.02 | 62.53 | 12.08 | 70.31 | 78.00 | 70.25 |
| Adapted OPT-1.3B with $\theta_p$ | ASPIRE ($\alpha = 0.0$) | 88.50 | 73.64 | 42.46 | 79.96 | 91.45 | 74.47 |
| | ASPIRE ($\alpha = 0.25$) | **90.76** | **79.26** | **44.03** | 83.06 | **93.41** | **81.17** |
| | ASPIRE ($\alpha = 0.5$) | 90.64 | 79.04 | 43.70 | **83.40** | 93.27 | 80.91 |
| | ASPIRE ($\alpha = 0.75$) | 89.84 | 76.58 | 42.05 | 80.97 | 92.96 | 79.90 |
| | ASPIRE ($\alpha = 1.0$) | 88.66 | 73.53 | 38.65 | 76.34 | 92.48 | 78.57 |
| Adapted OPT-2.7B with $\theta_p$ | ASPIRE ($\alpha = 0.0$) | 90.80 | 74.23 | 53.56 | 81.74 | 92.86 | 75.72 |
| | ASPIRE ($\alpha = 0.25$) | **92.63** | **80.25** | **55.06** | **84.44** | **94.73** | **82.60** |
| | ASPIRE ($\alpha = 0.5$) | 92.56 | 80.18 | 54.61 | 84.33 | 94.59 | 82.16 |
| | ASPIRE ($\alpha = 0.75$) | 92.05 | 78.37 | 52.71 | 81.52 | 94.28 | 80.98 |
| | ASPIRE ($\alpha = 1.0$) | 91.33 | 76.08 | 48.84 | 76.39 | 93.77 | 79.48 |

Table 17: Results of studying the effect of $\alpha$. All numbers are percentages. **Bold** numbers are superior results.

| Model | Method | CoQA | | TriviaQA | | SQuAD | |
|---|---|---|---|---|---|---|---|
| | | Acc ↑ | AUACC ↑ | Acc ↑ | AUACC ↑ | Acc ↑ | AUACC ↑ |
| Pre-trained OPT-350M | Self-Consistency | 24.54 | 37.83 | 3.41 | 7.93 | 5.75 | 15.59 |
| Adapted OPT-350M with $\theta_p$ | Self-Consistency | 59.03 | 74.09 | 7.41 | 18.40 | 63.87 | 79.99 |
| | ASPIRE (ours) | **59.46** | **75.55** | **8.25** | **19.00** | **64.74** | **82.59** |
| Pre-trained OPT-1.3B | Self-Consistency | 50.90 | 69.14 | 13.74 | 30.10 | 19.89 | 41.45 |
| Adapted OPT-1.3B with $\theta_p$ | Self-Consistency | 76.73 | 88.29 | 20.34 | 42.12 | 80.84 | 90.68 |
| | ASPIRE (ours) | **76.85** | **90.76** | **21.73** | **44.03** | **80.94** | **93.41** |
| Pre-trained OPT-2.7B | Self-Consistency | 57.60 | 75.27 | 19.57 | 39.88 | 23.73 | 47.78 |
| Adapted OPT-2.7B with $\theta_p$ | Self-Consistency | 80.41 | 90.61 | 27.59 | 52.20 | 83.11 | 92.34 |
| | ASPIRE (ours) | **80.45** | **92.63** | **29.21** | **55.06** | **83.27** | **94.73** |

Table 18: Comparing with self-consistency. All numbers are percentages. **Bold** numbers are superior results.