# OpenReview forum: "Adaptation with Self-Evaluation to Improve Selective Prediction in LLMs"
_EMNLP/2023/Conference — EMNLP 2023 Findings_

### Official Review · Reviewer_7xz9 · 2023-07-29

**Soundness:** 3

**Excitement:**

2: Mediocre: This paper makes marginal contributions (vs non-contemporaneous work), so I would rather not see it in the conference.

**Paper Topic And Main Contributions:**

This paper focuses on the selective prediction of language models, that is force the models to assess the accuracy of their predictions and refrain from making wrong predictions. The main contribution lies in the method of generating self-evaluate data and using it to tune the model for selective prediction.

**Reasons To Accept:**

1. The method proposed in this paper shows effectiveness when using OPT-2.7B and GPT2-XL, achieving the SOTA selective prediction performance on three question-answering datasets.

**Reasons To Reject:**

1. The authors claim that previous methods such as Self-consistency require multiple sampling, which is not effective. However, the method proposed in this paper requires two staging finetuning, and in addition, the method is task-specific, which means we have to tune the model for each task (and it also requires some training data), which results in a larger cost. In contrast, previous methods are more simple and effective.
2. Experiments in this paper are not sufficient for the claim of "improve selective prediction in LLMs", as only GPT2-XL and OPT-2.7B are adopted. I'm not sure about the effectiveness of this method truly LLMs (at least > 7B). Especially, for large models, they show self-evaluate ability without further tuning, so the method proposed in this paper seems unnecessary.
3. Some of the baselines, e.g. Self-eval might be effective on LLMs but weak on small models, so the comparison in table1 is not fair.

**Reproducibility:**

4: Could mostly reproduce the results, but there may be some variation because of sample variance or minor variations in their interpretation of the protocol or method.

**Reviewer Confidence:**

4: Quite sure. I tried to check the important points carefully. It's unlikely, though conceivable, that I missed something that should affect my ratings.

---

> ### Author Rebuttal · Authors · 2023-08-28
>
> Thank you for your valuable feedback, which has helped us improve our paper. We appreciate that you think our proposed method is “effective when using medium-size LLMs and achieves the SOTA selective prediction performance on three question answering datasets”.
>
> We address all your concerns and questions in detail below.
>
> **Question 1**: The authors claim that previous methods such as Self-consistency require multiple sampling, which is not effective. However, the method proposed in this paper requires two staging finetuning, and in addition, the method is task-specific, which means we have to tune the model for each task (and it also requires some training data), which results in a larger cost. In contrast, previous methods are more simple and effective.
>
> **Answer 1**: Although our approach introduces additional training cost, it is more effective compared to previous methods, including self-consistency, at test time. Our approach not only leads to better performance compared to previous methods, but also smaller inference costs, that can be on top of mind for many real-world use cases.
>
> **Question 2**: Experiments in this paper are not sufficient for the claim of "improve selective prediction in LLMs", as only GPT2-XL and OPT-2.7B are adopted. I'm not sure about the effectiveness of this method truly LLMs (at least > 7B). Especially, for large models, they show self-evaluation ability without further tuning, so the method proposed in this paper seems unnecessary.
>
> **Answer 2**: We don’t evaluate LLMs with the number of parameters greater than 7B due to computational constraints. However, the proposed approach can be applied to LLMs with any size if there are enough computational resources. Besides, we perform additional experiments to show that even for the pre-trained OPT model with 30B parameters, it still has limited self-evaluation ability (the AUROC of the Self-eval and P(True) methods are below 60%) on the three benchmarks we evaluate (see Table I). Thus, even for large LLMs with more than 7B parameters, further fine-tuning is still needed to boost their selective prediction performance on challenging question-answering datasets.
>
> Besides, the proposed method is useful for some practical applications where high selective prediction performance and low inference costs are desired after deploying the LLM. In those applications, practitioners prefer collecting some training data to fine-tune smaller LLMs to achieve high selective prediction performance than directly deploying very large pre-trained LLMs with limited selective prediction performance on the specific tasks.
>
>
> **Question 3**: Some of the baselines, e.g. Self-eval might be effective on LLMs but weak on small models, so the comparison in table1 is not fair.
>
> **Answer 3**: We perform experiments to compare the proposed ASPIRE with the adapted OPT-2.7B model and the baselines with the pre-trained OPT-30B model. The results in Table I show that  the proposed ASPIRE with the adapted OPT-2.7B model can significantly outperform the Self-eval and P(True) baselines with the pre-trained OPT-30B model in terms of the selective prediction performance (measured by AUACC and AUROC). Note that on the TriviaQA dataset, although the pre-trained OPT-30B model has better accuracy than the adapted OPT-2.7B model, the Self-eval and P(True) baselines with the pre-trained OPT-30B model have much worse selective prediction performance compared to the proposed ASPIRE with the adapted OPT-2.7B model. In summary, our results demonstrate that the self-eval approaches are not effective on LLMs with more than 7B parameters and applying the proposed ASPIRE to smaller LLMs can lead to better selective prediction performance compared to those self-eval approaches with much larger LLMs.
>
> **Table I**: *Comparing the proposed ASPIRE with the adapted OPT-2.7B model and the baselines with the pre-trained OPT-30B model. All numbers are percentages. Bold numbers are superior results.*
> | Dataset  | Model                            | Method        | Accuracy | AUACC   | AUROC  |
> |----------|----------------------------------|---------------|----------|---------|--------|
> | TriviaQA | Pre-trained OPT-30B              | Self-eval     | **39.36**   | 36.92  | 48.90 |
> | TriviaQA | Pre-trained OPT-30B              | P(True)       | **39.36**   | 36.20  | 45.63 |
> | TriviaQA | Adapted OPT-2.7B with $\theta_p$ | ASPIRE (ours) | 29.21   | **55.06**  | **84.44** |
> | CoQA     | Pre-trained OPT-30B              | Self-eval     | 71.06   | 71.99  | 51.10 |
> | CoQA     | Pre-trained OPT-30B              | P(True)       | 71.06   | 71.59  | 51.31 |
> | CoQA     | Adapted OPT-2.7B with $\theta_p$ | ASPIRE (ours) | **80.45**   | **92.63**  | **80.25** |
> | SQuAD    | Pre-trained OPT-30B              | Self-eval     | 41.41   | 46.24  | 57.26 |
> | SQuAD    | Adapted OPT-2.7B with $\theta_p$ | ASPIRE (ours) | **83.27**   | **94.73**  | **82.60** |

---

### Official Review · Reviewer_x6AJ · 2023-08-05

**Soundness:** 4

**Excitement:**

3: Ambivalent: It has merits (e.g., it reports state-of-the-art results, the idea is nice), but there are key weaknesses (e.g., it describes incremental work), and it can significantly benefit from another round of revision. However, I won't object to accepting it if my co-reviewers champion it.

**Paper Topic And Main Contributions:**

This paper studies uncertainty measurement for large language models, specifically the selective prediction for large language models, i.e., the model learns to refuse to answer questions that the model is uncertain about the prediction.

The paper proposes an effective method for the studied problem, which first tunes the model on the task-specific data, and then uses the tuned model to sample answers(some answers are wrong, some are correct), finally, trains a calibrator to classify the correct and wrong answers.

The experiments demonstrate the effectiveness of the proposed method.

**Questions For The Authors:**

1. The model has been fine-tuned with the training data, and then the fine-tuned model is used to sample answers to the questions from the training set. How you address the problem that the model may have memorize the gold answers?




**Reasons To Accept:**

1. The proposed method is simple and effective.

The proposed method is easy to follow and can be implemented easily in practice. The method is mainly about sampling different answers from the model, and these answers may contain noises, then learning a classifier to classify the correct and wrong answers.

2. The experiments are solid.

To verify the effectiveness of the method, the paper tested three QA datasets. According to the experimental results, the proposed method outperforms the baselines significantly.

The ablation study further explains how the proposed method works.

**Reasons To Reject:**

1. The method may lack novelty.

An essential part which is the M(y1, y2) function comes from a previous work (described in 204 ~ 213) (The previous work: Semantic uncertainty: Linguistic invariances for uncertainty estimation in natural language generation).

The newly proposed technics in this paper are only sampling from the model, and then training a classifier.

That's why I would think the novelty of the proposed method may be limited.

**Reproducibility:**

4: Could mostly reproduce the results, but there may be some variation because of sample variance or minor variations in their interpretation of the protocol or method.

**Reviewer Confidence:**

4: Quite sure. I tried to check the important points carefully. It's unlikely, though conceivable, that I missed something that should affect my ratings.

---

> ### Author Rebuttal · Authors · 2023-08-28
>
> Thank you for your valuable feedback, which has helped us improve our paper. We appreciate that you think our proposed method “is simple and effective, and the experiments are solid”.
>
> We address all your concerns and questions in detail below.
>
> **Question 1**: The method may lack novelty. An essential part which is the M(y1, y2) function comes from a previous work (described in 204 ~ 213) (The previous work: Semantic uncertainty: Linguistic invariances for uncertainty estimation in natural language generation). The newly proposed techniques in this paper are only sampling from the model, and then training a classifier.
>
> **Answer 1**: We indeed use the $M(y_1, y_2)$ function to evaluate the similarity of the generated output sequence compared to the reference output sequence. However, using $M(y_1, y_2)$ is not our contribution. Our contributions include proposing a novel training framework for learning self-evaluation to enhance selective prediction performance; and conducting extensive experiments on three question-answering benchmarks using various LLMs to demonstrate the effectiveness of the proposed method in improving selective prediction performance. Although the techniques may look simple, they are novel and effective in improving the selective prediction performance. Also, the proposed method has good training sample efficiency (see Table 5) and introduces low inference costs.
>
> **Question 2**: The model has been fine-tuned with the training data, and then the fine-tuned model is used to sample answers to the questions from the training set. How do you address the problem that the model may have memorized the gold answers?
>
> **Answer 2**: Even if the model memorizes the gold answer for a given question, we can still sample wrong answers from it for the given question (e.g., via beam search). Since LLMs generate the output sequence in an auto-regressive way, even if the likelihood on the gold answer for the given question is high, the likelihood on some wrong answers might also be high. In our experiments, we observe that we could find several high-likelihood wrong answers (e.g., 8 wrong answers out of 10 sampled answers) via beam search for some questions where the LLM could generate a correct answer with greedy-decoding. The proposed method can sample those wrong answers and fine-tune the LLM to distinguish correct and wrong answers.

---

### Official Review · Reviewer_1jxs · 2023-08-10

**Typos Grammar Style And Presentation Improvements:** 1. Should $k_c$ in line 316 and $k_w$…
**Soundness:** 3

**Excitement:**

3: Ambivalent: It has merits (e.g., it reports state-of-the-art results, the idea is nice), but there are key weaknesses (e.g., it describes incremental work), and it can significantly benefit from another round of revision. However, I won't object to accepting it if my co-reviewers champion it.

**Paper Topic And Main Contributions:**

The paper proposes the ASPIRE methodology for Question Answering tasks. The ASPIRE methodology prevents an LLM from generating token sequences when they aren't unsure of the answer. First, each example in the training set is used to generate multiple answers given the question. A metric is then used to determine if the generated answers are correct answers or incorrect answers. The model is then trained to increase the likelihood of positive answers and reduce the likelihood of negative answers. The methodology shows strong empirical performance on 3 benchmark dataset on 2 metrics.

**Questions For The Authors:**

See the reasons to reject the paper

**Reasons To Accept:**

1. The methodology is simple yet interesting and performs better than other competing baseline methodologies.
2. The authors have conducted an appropriate number of experiments on 3 benchmark datasets.

**Reasons To Reject:**

1. The description of the metrics is limited. it would be desirable to have an explanation of the metrics used in the paper. Or at least a citation to the metrics would have been good.
2. The training objective in Equation 7 would increase the likelihood of negative cases as well resulting in unwanted behavior. Should the objective be: \mathcal{L}_{c} - \mathcal{L}_{w}?
3. The paper needs a bit of polishing as at times equations are clubbed together. The equations in Sections 4 and 5 can be clubbed together while introducing them.
4. The paper motivates by the fact that we need to generate multiple sequences during the test time and progress to get rid of them. However, ASPIRE generates multiple answers during the training phase. This should be explicitly mentioned in the paper as it directly conflicts with the claim of not generating multiple sequences.
5. A bit more analysis on the impact of the number of model parameters is warranted.




**Reproducibility:**

3: Could reproduce the results with some difficulty. The settings of parameters are underspecified or subjectively determined; the training/evaluation data are not widely available.

**Reviewer Confidence:**

4: Quite sure. I tried to check the important points carefully. It's unlikely, though conceivable, that I missed something that should affect my ratings.

---

> ### Author Rebuttal · Authors · 2023-08-28
>
> Thank you for your valuable feedback, which has helped us improve our paper. We appreciate that you think our paper proposes “a simple yet interesting method that outperforms other competing baselines as demonstrated by extensive experiments”.
>
> We address all your concerns and questions in detail below.
>
> **Question 1**: The description of the metrics is limited. it would be desirable to have an explanation of the metrics used in the paper. Or at least a citation to the metrics would have been good.
>
> **Answer 1**: We use the area under the accuracy-coverage curve (AUACC) metric to measure selective prediction performance and use the area under the receiver operator characteristic curve (AUROC) metric to measure the quality of the selection score estimation. AUACC is the common metric used for evaluating selective prediction performance [1][2]. AUROC is equivalent to the probability that a randomly chosen correct output sequence has a higher selection score than a randomly chosen incorrect output sequence. AUROC is used in Kuhn et al. (2023) for evaluating uncertainty estimation methods. We also use the Rouge-L (Lin and Och, 2004) to evaluate the similarity of the generated answers with the reference answers, following Kuhn et al. (2023). We will clarify these.
>
> [1] Xin, Ji, et al. "The art of abstention: Selective prediction and error regularization for natural language processing." Proceedings of the 59th Annual Meeting of the Association for Computational Linguistics and the 11th International Joint Conference on Natural Language Processing (Volume 1: Long Papers). 2021.
>
> [2] Yoshikawa, Hiyori, and Naoaki Okazaki. "Selective-LAMA: Selective Prediction for Confidence-Aware Evaluation of Language Models." Findings of the Association for Computational Linguistics: EACL 2023. 2023.
>
> **Question 2**: The training objective in Equation 7 would increase the likelihood of negative cases as well resulting in unwanted behavior. Should the objective be: \mathcal{L}{c} - \mathcal{L}{w}?
>
> **Answer 2**: The original training objective (Equation 7) is indeed correct. Given the input query $\textbf{x}$, for those wrong output sequences, we would like to maximize the probability of the “incorrect” token so that the probability of the “correct” token would be minimized. This can result in a low learned self-eval score for the incorrect output sequences, encouraging better selective prediction performance.
>
> **Question 3**: The paper needs a bit of polishing as at times equations are clubbed together. The equations in Sections 4 and 5 can be clubbed together while introducing them.
>
> **Answer 3**: We will polish Sections 4 and 5 to make the equations separate.
>
> **Question 4**: The paper is motivated by the fact that we need to generate multiple sequences during the test time and progress to get rid of them. However, ASPIRE generates multiple answers during the training phase. This should be explicitly mentioned in the paper as it directly conflicts with the claim of not generating multiple sequences.
>
> **Answer 4**: Yes, during the training phase, ASPIRE needs to generate multiple answers. However, it doesn’t conflict with our claim, as our claim is that ASPIRE doesn’t need to generate multiple sequences **at the test time**. Due to this, the proposed ASPIRE has smaller inference costs compared to previous methods.
>
>
> **Question 5**: A bit more analysis on the impact of the number of model parameters is warranted.
>
> **Answer 5**: Larger number of model parameters can lead to better prediction performance. However, it would also lead to higher training and inference costs. Besides, we show that even if the number of model parameters is very large (e.g. using the OPT-30B model. Refer to Answer 2 to the reviewer 7xz9 for details), the model still has limited self-evaluation performance. Thus, the proposed framework is useful for adapting LLMs to learn self-evaluation across various sizes of LLMs.
>
> **Question 6**: Should $k_c$ in line 316 and $k_w$ in line 319 be $z_c$ and $z_w$ respectively?
>
> **Answer 6**: $k_c$ and $k_w$ represent the number of correct / wrong outputs while $z_c$ and $z_w$ are a pair of tokens that represent the words “correct” and “incorrect” respectively.

---

### Meta-Review · Area_Chair_7YQ6 · 2023-09-22

**Recommendation:** 3

**Metareview:**

This work proposes a selective prediction technique that discourages LLMs from predicting an answer when unsure. Towards this goal, an LLM is fine-tuned on task-specific data and multiple answers are sampled (some correct, some wrong) from the fine-tuned model. A classifier is trained to discriminate between correct and wrong answers, and the model is tuned to refrain from predicting wrong answers. The model incurs low inference costs and is also fairly sample-efficient during training.

All the reviewers have rated this work as good or higher on soundness. Concerns about whether the proposed technique is effective
for larger LLMs was addressed by the authors in their rebuttal. While inference costs are minimal and this is a desired feature in many use-cases, training efficiency does take a bit of a hit using the proposed technique with sampling multiple answers and task-specific finetuning of the model.

---

### Decision · Program_Chairs · 2023-10-07

**Decision:**

Accept-Findings

**Comment:**

This work proposes a selective prediction technique that discourages LLMs from predicting an answer when unsure. Towards this goal, an LLM is fine-tuned on task-specific data and multiple answers are sampled (some correct, some wrong) from the fine-tuned model. A classifier is trained to discriminate between correct and wrong answers, and the model is tuned to refrain from predicting wrong answers. The model incurs low inference costs and is also fairly sample-efficient during training.

All the reviewers have rated this work as good or higher on soundness. Concerns about whether the proposed technique is effective
for larger LLMs was addressed by the authors in their rebuttal. While inference costs are minimal and this is a desired feature in many use-cases, training efficiency does take a bit of a hit using the proposed technique with sampling multiple answers and task-specific finetuning of the model.